Hybrid transformer-CNN and LSTM model for lung disease segmentation and classification

Shafi Syed Mohammed
Chinnappan Sathiya Kumar sathiyakumar.c@vit.ac.in
Vellore Institute of Technology , Vellore, Tamil Nadu , India
Alatas Bilal
Electronic publication date: 2024 Dec 13
Publication date: 2024
Volume: 10
Electronic Location ID: e2444
Received 2024 May 22; Accepted 2024 Oct 1
Copyright: © 2024 Shafi and Chinnappan
Copyright year: 2024
Copyright holder: Shafi and Chinnappan
License: This is an open access article distributed under the terms of the Creative Commons Attribution License, which permits unrestricted use, distribution, reproduction and adaptation in any medium and for any purpose provided that it is properly attributed. For attribution, the original author(s), title, publication source (PeerJ Computer Science) and either DOI or URL of the article must be cited.
License URL: https://creativecommons.org/licenses/by/4.0/

Keywords: Classification, L-MLSTM, Lung disease, Median filtering, Modified LGIP, Segmentation

Funding: The authors received no funding for this work.

==============================
According to the World Health Organization (WHO) report, lung disorders are the third leading cause of mortality worldwide. Approximately three million individuals are affected with various types of lung disorders annually. This issue alarms us to take control measures related to early diagnostics, accurate treatment procedures, etc. The precise identification through the assessment of medical images is crucial for pulmonary disease diagnosis. Also, it remains a formidable challenge due to the diverse and unpredictable nature of pathological lung appearances and shapes. Therefore, the efficient lung disease segmentation and classification model is essential. By taking this initiative, a novel lung disease segmentation with a hybrid LinkNet-Modified LSTM (L-MLSTM) model is proposed in this research article. The proposed model utilizes four essential and fundamental steps for its implementation. The first step is pre-processing, where the input lung images are pre-processed using median filtering. Consequently, an improved Transformer-based convolutional neural network (CNN) model (ITCNN) is proposed to segment the affected region in the segmentation process. After segmentation, essential features such as texture, shape, color, and deep features are retrieved. Specifically, texture features are extracted using modified Local Gradient Increasing Pattern (LGIP) and Multi-texton analysis. Then, the classification step utilizes a hybrid model, the L-MLSTM model. This work leverages two datasets such as the COVID-19 normal pneumonia-CT images dataset (Dataset 1) and the Chest CT scan images dataset (Dataset 2). The dataset is crucial for training and evaluating the model, providing a comprehensive basis for robust and generalizable results. The L-MLSTM model outperforms several existing models, including HDE-NN, DBN, LSTM, LINKNET, SVM, Bi-GRU, RNN, CNN, and VGG19 + CNN, with accuracies of 89% and 95% at learning percentages of 70 and 90, respectively, for datasets 1 and 2. The improved accuracy achieved by the L-MLSTM model highlights its capability to better handle the complexity and variability in lung images. This hybrid approach enhances the model’s ability to distinguish between different types of lung diseases and reduces diagnostic errors compared to existing methods.

Introduction

About 3 million people are affected by various kinds of lung disorders that cause death in humans, and it is the third prominent death caused by infectious disease, as per a report by WHO (Kim et al., 2022). Air pollution, smoking, and other genomic factors are the causes of lung disorders. If the lungs are affected by severe infection without medication, it leads to death (Ferl et al., 2022). Recently, severe acute respiratory syndrome coronavirus 2 (SARS-CoV-2) has been revealed as a highly infectious human disease that specifically targets the lungs and can cause severe harm (Karar et al., 2023).

Advanced clinical diagnosis systems using Artificial Intelligence (AI) techniques aid early detection of lung infections before severe damage occurs in the lungs (Wang et al., 2023). Recently, there has been a significant demand for developing intelligent computerized clinical diagnosis systems (CCDS) utilizing AI methods to assist medical professionals in achieving precise clinical diagnosis and treatment (Han et al., 2020) computerized tomography (CT), magnetic resonance imaging (MRI), and chest X-Ray (CXR) are the medical imaging tools that play an essential role in CCDS for diagnosing lung disease (Ozdemir, Russell & Berlin, 2020). They give more valuable data about lung infection with higher accuracy and precision (Bartoli et al., 2022). Such image models are beneficial for lung disease diagnosis (Maleki & Niaki, 2023). CT gives more accurate and complete lung information than CXR (Wang et al., 2021). The lung image can be segmented from a CT scan or CXR to develop CCDS (Müller, Soto-Rey & Kramer, 2021). Therefore, the classification and segmentation process is essential.

Lung segmentation is a computer-based method to detect the boundaries of lungs from surrounding thoracic tissue on CT images (Medeiros et al., 2019; Linning et al., 2019). Numerous lung segmentation algorithms are proposed using conventional techniques such as thresholding, edge-based, region-based, water-shed, local binary pattern (LBP), and Gray-Level Co-Occurrence Matrix (GLCM) (Amyar et al., 2020; Yang et al., 2023b). The classification of lung diseases provides in-depth information to realize a patient management perspective for different types of lung diseases that would be needed for various treatment methods (Khanna, Londhe & Semwal, 2020; Halder et al., 2020). Conventional methods of classifying lung disease include the visual interpretation of imaging studies (Nanglia et al., 2021). However, this technique is subjective and may result in inter-observer variability (Hu et al., 2020).

Lung diseases, including conditions such as COVID-19 and pneumonia, pose significant challenges to healthcare systems worldwide. Accurate and timely diagnosis is crucial for effective treatment and management of these diseases. Traditional methods of diagnosis, such as manual examination of chest CT scans, often suffer from limitations related to variability in interpretation and high reliance on the expertise of radiologists. As the volume of medical imaging data grows, there is an increasing need for automated and reliable diagnostic tools that can assist in the early detection and accurate classification of lung diseases. Recent advancements in machine learning and computer vision have shown promise in enhancing medical image analysis. However, existing methods often face challenges such as insufficient segmentation accuracy, limited feature extraction capabilities, and inefficiencies in handling complex patterns in medical images. To address these issues, this research proposes a novel approach for automatically examining medical images for classifying and segmenting lung diseases. By integrating advanced techniques such as Improved Transformer-based convolutional neural network (ITCNN) model for segmentation, modified Local Gradient Increasing Pattern (LGIP) and Multi-texton analysis for feature extraction and hybrid LinkNet-Modified LSTM (L-MLSTM) our approach aims to address the limitations of existing methods, providing a more accurate, reliable, and efficient solution for lung disease diagnosis. This integration ensures that our model can handle the complexities of lung disease diagnostics with greater precision and efficacy. This comprehensive approach not only enhances segmentation and classification but also contributes to the broader goal of improving patient outcomes through advanced diagnostic tools. The unique combination of improved segmentation, sophisticated feature extraction, and hybrid classification in our L-MLSTM model distinguishes it from existing methods, making it a significant contribution to the field of medical image analysis and lung disease diagnosis.

The proposed model’s effectiveness is enhanced with some improvements, briefly explained below. Introduces an improved Transformer-based convolutional neural network (CNN) model for segmenting diseased lung regions, integrating a modified Swin Transformer block with structural enhancements, including batch normalization (BN) and average pooling layers. These innovations significantly enhance the model’s ability to precisely identify and delineate pathological regions in lung images.

Develops a modified Local Gradient Increasing Pattern (LGIP) technique for local texture feature extraction, capturing detailed regional gradient variations in segmented images more effectively than conventional methods to provide more comprehensive and informative texture data crucial for accurate disease classification.

Proposes the LinkNet-Modified LSTM (L-MLSTM) model, which combines the LinkNet architecture with a modified long-short term memory (MLSTM) network. The model utilizes a novel ScReLU activation function, integrating Sigmoid, comb-H-sine, and Leaky ReLU functions, to enhance classification performance and reliable detection and classification of lung diseases.

This research article on proposed lung disease segmentation with a hybrid L-MLSTM classification model comprises these improvements. “Literature review” reviews existing techniques, “ Methodology of Lung Disease Segmentation with Hybrid L-MLSTM Classification Model” explains the proposed model’s procedural process, and “Result and Discussion” experiments with the proposed model under several analyses. Finally, “Conclusion” concludes the article.

Literature review

In 2022, Salama & Aly (2022) proposed a framework for segmenting, classifying, and identifying CT image features to determine COVID-19 status based on lung CT images. Weiner’s filter and the Image Size Dependent Normalization Technique (ISDNT) use pre-processing techniques to enhance images and reduce noise. To address the lack of COVID-19 CT lung images and reduce the possibility of overfitting in deep learning (DL) models, a mix of transfer learning (TL) and data augmentation techniques was used. However, this framework often struggles to fully utilize multi-scale context information in CT scans. This limitation affects the accuracy of segmenting COVID-19-infected areas. It focuses on specific types of CT images or dataset characteristics, which could limit the generalizability of the framework.

In 2023, Punitha et al. (2023) introduced a computer-aided diagnosis (CAD) system for COVID-19 that identified and classified abnormalities in lung CT images using Artificial Bee Colony optimized ANN (ABCNN). Using an Artificial Bee Colony (ABC)-optimized region-growing process, this method isolated sick regions from CT images of both normal and COVID patients. Moreover, ABC optimization was used to fine-tune an optimal artificial neural network (ANN) model that included input data, starting weights, and hidden nodes in order to categorize these aberrant regions into COVID and normal labels. However, the performance of ANN heavily relies on the quality and diversity of the training data. If the training dataset lacks diversity or does not represent the full spectrum of COVID-19 cases, the model may struggle to generalize well to unseen data. This limitation could affect the ANN’s ability to accurately detect COVID-19 in real-world scenarios.

In 2023, Raza et al. (2023) introduced a new transfer learning-based predictor known as Lung-EffNet to classify lung cancer. The Efficient Net model was modified and adapted to create Lung-EffNet by including additional top layers. The standardized “Iraq-Oncology Teaching Hospital/National Center for Cancer Diseases (IQ-OTH/NCCD)” dataset was used for experimental evaluations, in which medical images of patients with lung cancer were categorized according to whether they contained benign, malignant, or normal lung nodules. However, EfficientNet models are computationally intensive, especially during training. Deploying such models in clinical settings might require substantial computational resources, which could be a limitation for smaller institutions. Moreover, it focuses solely on lung cancer classification without addressing the model’s performance on other lung diseases or its adaptability to various pathological conditions.

In 2022, Alshayeji, Chandrabhasi Sindhu & Abed (2022) proposed a CAD system to distinguish COVID-19 patients from normal cases. Using CT scans, the infection region segmentation procedure and infection severity estimate can be carried out. For COVID-19, this model produced an accurate and fully automated real-time CAD framework. However, CAD systems often lack transparency in their decision-making process. Understanding why the system makes specific predictions (e.g., identifying infected regions) is crucial for clinical acceptance. It focuses on specific stages or types of COVID-19, which could limit the model’s effectiveness across different disease variants or new strains.

In 2021, Fung et al. (2021) introduced a self-supervised two-stage deep-learning model to segment COVID-19 lesions from chest CT images to support rapid COVID-19 diagnosis. Two real-world datasets were used to assess the model’s performance compared to the earlier technique. The clinical and biological pathways were examined using features engineered from the anticipated lung lesions. However, self-supervised deep learning models can learn useful features from unlabeled data, their effectiveness depends on the diversity of cases encountered during training. It has not addressed how the model performs over time or under varying disease progression scenarios, which are crucial for understanding disease dynamics.

In 2020, Liu et al. (2020) used deep convolutional network algorithms, multi-scale superpixels, and random forest (RF) to accurately segment pathological lungs from thoracic CT images. Using multi-scale superpixels, the algorithm segments the diseased image, extracts features, and applies group RF classifiers. To get an initial segmentation, it integrates the classification results of RFs using a fractional-order grey correlation approach. Using a divide-and-conquer approach, the program then refines the left and right lung segmentation. The algorithm’s decision-making process lacks transparency, especially the fusion of RF classifiers and the divide-and-conquer strategy. Understanding the need for specific regions to be segmented or repaired is essential for clinical acceptance. A focus on static images without addressing temporal changes or disease progression could be a limitation. Limited evaluation across diverse datasets and disease variants might affect the model’s applicability and reliability.

In 2022, Harsono, Liawatimena & Cenggoro (2022) proposed a new lung nodule detection and classification model using an Inflated 3D ConvNet Retina Net (I3DR-Net) stage detector. The pre-trained weights from natural photos were combined to create the model. I3DR-Net, which was developed using the I3D backbone and a feature pyramid network optimized for a multi-scale 3D thorax CT-scan dataset, showed superior performance in identifying and categorizing malignant nodules when compared to the previous top models, UFRCNN + mAP and Retina U-Net. Lung nodules exhibit diverse shapes, sizes, and textures. Some are subtle and challenging to distinguish from normal lung tissue. The model must handle this variability effectively. The focus on lung nodule detection may limit the model’s generalizability to other types of lung diseases or pathological conditions, potentially affecting its versatility.

In 2023, Murugappan et al. (2023) proposed an automated segmented model using CT images and a unified DeepLabV3+ network to identify four regions in COVID-19 patients. The proposed DeepLabV3+ network is designed for efficient lung segmentation using chest CT-scan images by selecting the pre-trained network, image sizes, network hyperparameter selection, and computation time. The four-class segmentation of chest CT scans can be used to identify the affected lung region and develop an automated computerized clinical diagnosis system for COVID-19. However, DeepLabV3+ networks are computationally intensive. Deploying such models in clinical settings may require substantial resources. It does not address the temporal aspects of disease progression or changes over time, which are crucial for monitoring and predicting disease evolution.

In 2020, Demir, Sengur & Bajaj (2020) proposed a disease classification model via lung sound signals using the International Conference on Biomedical Health Informatics (ICBHI) 2017 dataset. Initially, the signals in this dataset were processed by short time Fourier transform (STFT). Then, deep CNN with TL was used for feature extraction and support vector machine (SVM) for classification in the lung sound classification phase. However, efficient approaches often rely on transfer learning or pre-trained models. Ensuring robust generalization to unseen patients, scanners, and imaging protocols is essential.

In 2023, Alshmrani et al. (2023) developed a deep learning (DL)-based multi-class classification model to recognize and classify tuberculosis (TB), pneumonia, lung opacity, lung cancer, and COVID-19. The dataset with CXR images was resized, normalized, and arbitrarily segmented for further analysis by the DL model. Visual Geometry Group (VGG19) and CNN combined models were used for retrieving features and disease classification. The presence of imbalanced classes in the CXR dataset can affect model performance. Some lung diseases may be rare, leading to insufficient samples for effective training. The focus on lung disease classification does not encompass a wide range of diseases or conditions, limiting the model’s versatility.

In 2022, Yi et al. (2022) proposed a Residual Encoder-Decoder Convolutional Neural Network (RED-CNN) deep-learning model for multi-classification of five pulmonary diseases. The RED-CNN is a brand-new network built using the RED block and based on the CNN architecture. The Res2Net, ECA, and Double BlazeBlock modules comprise the RED block. Together, they can extract more detailed data, provide cross-channel information, and improve the extraction of global data with robust feature extraction capabilities. However, the availability of comprehensive datasets containing samples of all the mentioned diseases is challenging. It does not thoroughly evaluate the model’s performance across different variants of pulmonary diseases or under diverse clinical conditions.

In 2023, Yi et al. (2023) developed a cellular apoptosis susceptibility gene (CAS) breast cancer diagnosis framework. Initially, a publicly available dataset will be used to train the segmentation network CRA-ENet using masks. Subsequently, the weights are applied to segment non-mask clinical data to derive masks. Afterwards, the masks should be covered with the matching original pictures so that the ultrasound images only show the area affected by the lesion. Lastly, pictures are processed using SA-Net to distinguish between benign and malignant tumors. While CAS is associated with apoptosis, its precise role can vary depending on the cellular context. Depending on the specific conditions and interactions with other proteins, it may act as an apoptosis inhibitor or facilitator. There is a gap in addressing how the framework handles temporal changes or long-term monitoring of lesions.

In 2024, Gugulothu & Balaji (2024) has developed a hybrid deep-learning approach combining CNNs and LSTMs for early lung nodule detection from CT images. CNN networks are used to extract features from CT images. CNNs are known for their ability to learn spatial hierarchies of features through convolutional layers, making them effective for image-based tasks. LSTM networks are employed to capture temporal patterns and dependencies in the features extracted by the CNNs. LSTMs are particularly useful in handling sequences and temporal data. The CNN and LSTM components are integrated to leverage both spatial and temporal information for lung nodule classification. This hybrid approach aims to enhance the model’s accuracy and robustness for early nodule detection and classification. The model effectively integrates spatial and temporal information, but it faces challenges related to computational demands and dataset variability.

In 2023, Manickavasagam & Sugumaran (2023) has focused on optimizing a deep belief network for lung cancer detection and survival rate prediction. The approach enhances diagnostic accuracy and prognostic predictions, though it involves complex model optimization and is dependent on high-quality data. The study emphasizes optimizing the DBN configuration, including hyperparameters and network architecture, to improve detection accuracy and survival predictions. The DBN model is used to identify lung cancer from medical images and predict patient survival rates based on various features extracted from the data. The optimized DBN showed significant improvements in detecting lung cancer and predicting survival rates. The optimization process enhanced the model’s accuracy and robustness, providing valuable insights into patient prognosis. However, it is computationally intensive for training purposes. It does not fully explore how these predictions can be integrated with treatment planning or personalized medicine strategies.

Research gaps

Existing models for lung disease segmentation and classification, such as traditional CNNs, LSTM, and even advanced architectures like VGG19 + CNN, often face challenges in achieving high accuracy and generalizability. Many of these models struggle with accurately identifying and segmenting disease regions due to their limited ability to handle diverse and complex image features. The problem faced around the growth and application of DL methods for classifying and segmenting lung diseases in medical imaging of CXRs and CT scans. These problems can involve the accurate definition of lung structures and the determination of several pathological conditions. Some limitations are addressed: the potential overfitting of deep learning models limited annotated data, and data imbalance among disease classes in Table 1. Traditional approaches are not effectively integrating advanced techniques like Transformer-based methods and sophisticated feature extraction. This can result in suboptimal performance, especially when dealing with diverse lung disease datasets. Existing methods rely on basic feature extraction techniques that fail to capture the intricate details of lung diseases. For example, traditional methods do not fully exploit texture, shape, color, and deep features, leading to less accurate disease characterization. Models such as standard CNNs and LSTMs do not effectively address the sequential dependencies in image data, which can be critical for capturing the progression or subtle variations in lung diseases. To address the limitations of existing models, our research introduces the hybrid LinkNet-Modified LSTM (L-MLSTM) model. This model combines advanced segmentation with an improved Transformer-based CNN (ITCNN) to capture intricate features and provide accurate segmentation of affected lung regions. The L-MLSTM model is designed to enhance accuracy and robustness by integrating these advanced techniques. Our approach incorporates sophisticated feature extraction methods, including modified Local Gradient Increasing Pattern (LGIP) and Multi-texton analysis, to thoroughly analyze texture, shape, color, and deep features. This comprehensive feature extraction ensures that subtle and complex disease characteristics are captured, leading to improved classification performance. By leveraging diverse datasets (COVID-19 normal pneumonia CT images and chest CT scans), the L-MLSTM model is trained to handle a wide range of lung disease scenarios. This helps in improving the generalizability of the model, making it more effective across different types of lung diseases and enhancing its applicability in clinical settings.

Table 1 Features and limitations in existing lung disease-related techniques.

Author (Citation)	Methodology	Features	Limitations	Dataset used	
Salama & Aly (2022)	U-Net	The proposed method of data augmentation and TL steps are used to improve performance classification.	However, it is essential for high computational time limits in real-time applications.	COVID-19 lung CT images dataset	
Punitha et al. (2023)	ABCNN	This approach is characterized by its simplicity, achieved through a wrapper-based method, and it is also influential in determining optimal parameters for a precise segmentation and classification process.	It is essential to have high-dimensional COVID-19 datasets with efficient DL models for accurate results.	COVID-19 CT lung images dataset	
Raza et al. (2023)	Lung-EffNet	This model achieved a precision rate of 99.10% and robust ROC scores ranging from 0.97 to 0.99 on test sets, surpassing other CNN models.	Expanding the dataset would confirm the proposed method's effectiveness.	IQ-OTH/NCCD dataset	
Alshayeji, Chandrabhasi Sindhu & Abed (2022)	CAD-based COVID-19 disease classification and segmentation	It attained better results during the segmentation process because of the efficient segmentation technique, evaluated in terms of mean BF and a weighted IoU.	However, using a hybrid classification model to classify disease stages would attain better results.	COVID-19 CT lung images dataset	
Fung et al. (2021)	SSInfNet with baseline models	Sensitivity analysis of the proposed method demonstrated its robustness and generalizability.	It required significant computational resources, including powerful GPUs and substantial memory. This could be a limitation for real-time healthcare institutions with limited resources.	COVID-19 CT lung images	
Liu et al. (2020)	RF	This method efficiently segmented images using a divide-and-conquer scheme, enhancing segmentation accuracy.	It did not have enhanced fusion methods or augmented training samples to improve its performance.	Pathological thoracic CT images	
Harsono, Liawatimena & Cenggoro (2022)	13DR-Net	This model used natural image datasets and attained reduced training time, eventually increasing its performance.	It could not be applied to real-time applications since it did not integrate with a suitable software interface, cloud computing, and GPU.	LIDC-IDRI dataset	
Murugappan et al. (2023)	Deep neural network (DNN)	It provides a fully automated solution for lung segmentation, reducing the need for manual intervention and increasing efficiency.	Training deep neural networks requires significant computational resources, including GPUs and large amounts of memory, which might not be accessible to all practitioners.	LIDC-IDRI dataset	
Amyar et al. (2020)	Multi-task deep learning model	The model segments regions of interest (ROIs) in the CT images, particularly lung regions affected by pneumonia. It provides detailed spatial localization of the disease within the lung tissue.	The multi-task learning approach increases the complexity of the model, leading to higher computational requirements and longer training times	CT imaging data	
Demir, Sengur & Bajaj (2020)	Convolutional neural network (CNN) model	It optimizes the CNN architecture to achieve high classification performance with reduced computational resources.	Additional validation on external datasets is necessary to assess generalizability	Lung image dataset	
Alshmrani et al. (2023)	Deep learning architecture	It extracts features from the CXR images, capturing relevant patterns and abnormalities	If the dataset lacks sufficient variation or examples of certain disease types, the model’s ability to generalize is compromised.	Chest X-Ray (CXR) images	
Yi et al. (2022)	RED-CNN architecture	It classifies lung images into multiple categories of pulmonary diseases, leveraging the combined strength of convolutional and recurrent layers	The integration of recurrent layers with convolutional layers increases the computational complexity and training time of the model.	Pulmonary disease images	
Yi et al. (2023)	Cancerous area segmentation framework	It emphasizes the accurate identification and segmentation of lesion regions in ultrasound images	Inadequate data impact the model’s ability to generalize across different patient populations.	Ultrasound breast images	
Gugulothu & Balaji (2024)	Hybrid CNN + LSTM	It emphasizes early detection and classification of lung nodules to facilitate timely intervention.	The hybrid model is computationally intensive, requiring significant resources for training and inference.	Lung image database consortium image collection (LIDC-IDRI) dataset	
Manickavasagam & Sugumaran (2023)	Optimized DBN	It provides predictions related to patient survival, integrating diagnostic and prognostic aspects	The model’s performance is highly dependent on the quality and comprehensiveness of the dataset. Variations in data quality or completeness have affected the results.	Lung cancer dataset	

Methodology of lung disease segmentation with hybrid l-mlstm classification model

Lung diseases are a diverse category of disorders that affect the lungs and their ability to function correctly. These conditions can affect the airways, the lung tissue, or blood circulation through the lungs. Some common lung diseases include asthma, chronic obstructive pulmonary disease (COPD), pneumonia, lung cancer, etc. It significantly worsens the overall condition of the body, which leads to a higher risk of complications and even death. Without proper diagnosis and appropriate treatment by healthcare professionals, it severely affects the health of humans and may lead to life-threatening complications. So, segmentation and classification of lung diseases are vital for accurately detecting, diagnosing, and recognizing infected regions based on specific patterns, features, or characteristics, leading to identifying different lung disease types. This research article illustrates the steps followed by the proposed model’s pre-processing, segmentation, feature extraction, and classification stages, as shown in Fig. 1. The techniques performed in these steps are briefly listed below and descriptively explained in the following section. 1) Pre-processing is based on the median filtering technique.

2) Segmentation is based on the improved transformer-based CNN model.

3) Feature extraction is based on extracting texture features, shape features, color features, and deep features

4) Classification based on the hybrid L-MLSTM model.

Figure 1 Architectural flow of proposed lung disease segmentation classification model.

Pre-processing by median filtering

The median filter is a non-linear technique that removes noise and preserves the image’s edge. It works by taking the median value of the neighbourhood pixels. The median is obtained by sorting the pixels and selecting the middle value as the output pixel. It is an effective method to preserve the boundaries of an image, and its brightness is also conserved (Sreejith & Nayak, 2020). Median filtering is robust against outliers and extreme pixel values because it replaces each pixel’s value with the median of its neighbors, reducing the influence of unusual or extreme values. median filtering maintains sharpness and clarity of edges and fine details, which is crucial for tasks like medical image analysis where edge integrity is important. The application of the median filtering technique in the input image is carried out by following fundamental steps, which are listed below. Let us consider every pixel in the input image for pre-processing.

A sliding window (or kernel) of a specific size (e.g., 3 × 3, 5 × 5) is moved across the image. The size of the window determines how many neighboring pixels are considered in the filtering process.

For each position of the window, extract the pixel values within the window area.

Based on the intensities of pixels, the pixels are sorted in ascending or descending order to determine the median value. Sort the extracted pixel values and compute the median value. The median is the middle value in the sorted list if the number of pixels is odd, or the average of the two middle values if even.

Replace the central pixel value of the window with the computed median value.

Slide the window to the next position and repeat the process until all pixels in the image have been processed.

This process effectively reduces noise while preserving important image features, making median filtering a valuable technique in image preprocessing, especially in medical imaging. The resultant pre-processed image from the median filtering technique is obtained, which is indicated as Pk.

Segmentation by improved Transformer-based CNN

Segmentation is one of the fundamental processes in computer vision that involves partitioning an image into multiple segments to simplify its representation and facilitate more expressive analysis. Segmentation aims to identify and delineate diseased regions within the image. Here, an improved Transformer-based CNN model is proposed for image segmentation. Here, the process of segmentation is done on a pre-processed image.

Improved Transformer-based CNN model

In this improved Transformer-based CNN model (Fig. 2), segmentation on the pre-processed image is done by performing a feature retrieval technique in its blocks.

Figure 2 Architecture of improved Transformer-based CNN model for segmentation.

Generally, this model comprises three blocks: Encoder, Decoder, and Skip connection (Sun, Pang & Li, 2023). Here, the encoder block in the improved Transformer-based CNN model consists of two parallel branches: (i) Transformer branch and (ii) CNN branch (Yang et al., 2023a; Peng, Zhang & Guo, 2023). Each branch consists of a specific set of blocks, i.e., the Transformer branch has improved Swin Transformer blocks, and the CNN branch has Efficient Net blocks while these blocks perform the same process, ‘feature retrieval. The features retrieved by these blocks are then forwarded to the Swin Transformer and CNN Fusion Module (STCF), which performs convolutional operations to fuse those features. Finally, the resultant fused feature maps are obtained from each STCF module, which is then given to the decode block and skip connection to perform segmentation. Therefore, this model initially shows the pre-processed image as input into the Transformer and CNN branches. The improved Swin Transformer (IST) block performs feature retrieval in the Transformer branch on pre-processed images. Likewise, the CNN branch performs feature retrieval on the pre-processed images by Efficient Net block.

Improved Swin Transformer block in Transformer branch

The Swin Transformer is more efficient than standard Multi-head Self Attention (MSA) regarding data efficiency, computational complexity, etc. In contrast, the Swin Transformer is inefficient due to its lack of interpretability (Sun, Pang & Li, 2023). Therefore, the Transformer branch in the improved Transformer-based CNN model utilized an IST block for effectively retrieving features from pre-processed images with enhanced interpretability. This improved Transformer-based CNN model uses a conventional Swin Transformer with a Windows Multi-head Self Attention (W-MSA) module, Shifted Windows Multi-head Self Attention (SW-MSA) module, Layer Normalization (LN) layer, and Multilayer Perceptron (MLP) (Sun, Pang & Li, 2023). It is improved by adding a Batch Normalization (BN) layer and average pooling layer to improve its performance with enhanced interpretability. In addition, the W-MSA and SW-MSA modules in the IST block have improved self-attention compared to conventional ones. The w-MSA module works on pre-processed images, caps P to the k, and partitions it into several non-overlapping windows. An LN layer is also positioned before each W-MSA module, SW-MSA module, and MLP. MSA module combines with a moving window, a two-layer MLP with Gaussian Error Linear Unit (GELU) non-linearity, and a residual connection. Moreover, the Multi-Head Self-Attention in the W-MSA module executes only within each window, which reliably isolates the transferring of information between several windows and minimizes computational complexity. While using moving window operations, the SW-MSA module allows the exchanging of data between adjacent windows.

Moreover, the additional layers added in the IST block are described below. The BN layer is used to enhance the performance of the IST block by stabilizing and accelerating the training process by normalizing the input of each layer to have a standard normal distribution. Here, an average pooling layer is used to reduce the spatial dimensions of the pre-processed image, cap P to the k, and minimize the computations within the network. Further, this intermediate pooling layer controls overfitting issues and enhances the retrieval of robust features from pre-processed images. In the IST block, the modules such as W-MSA and SW-MSA are positioned in successive blocks using a window division mechanism. The formulation to express consecutive IST blocks using this mechanism is represented from Eqs. (1)–(4). Here, ui and u^i represent the output features of (S)W-MSA module and MLP in ith layer, respectively; W−MSA and SW−MSA are window-based multi-head self-attention mechanisms that employ standard window partitioning and shifted window partitioning approach (Sun, Pang & Li, 2023).

(1) u^i=W−MSA(LN(ui−1)+ui−1)

(2) ui=MLP(LN(u^i))+u^i

(3) u^i+1=SW−MSA(LN(ui)+ui)

(4) ui+1=MLP(LN(u^i+1))+u^i+1

Further, the improved self-attention mechanism in W-MSA and SW-MSA modules is the same. It is mathematically formulated as in Eq. (5). Here, q represents the query, k represents the key, v represents the value matrices. B represents the relative position parameter, introduced like the position embedding in Transformers. The variable D denotes the size of the dimension for each head. In contrast, the formulation of the conventional self-attention mechanism is represented in Eq. (6) wherein the terms νandkT(Transposeofkmatrix) denotes value and k fundamental matrices, respectively (Chen et al., 2023), where it is all values belong to Rp2×D (denotes the dimension of a query). The terms q∈Rp2×D,σandB′ indicates query matrix, standard deviation, and relative position bias parameter where its value is taken from bias matrix, B^∈R(2p−1)×(2p+1).

(5) ISA(q,k,ν)=Softmax[σ((qk∧T×qk)/√D+B)ν]

(6) SA(q,k,ν)=Softmax((qk∧T)/√D+B)ν

Efficient net block in CNN branch

Generally, an Efficient Net block is developed by neural architecture search, which comprises the ‘n’ number of Mobile Inverted Bottleneck Convolution (MBConvBlock). Each MBConvBlock comprises five essential components: the Convolution layer, BN layer, Squeeze & Excitation (SE) block, dropout layer, and Swish activation layer. SE block involves layers such as the global average pooling layer and two individual fully connected layers. Compound scaling is incorporated in the Efficient Net block for enhanced performance even with the same number of attributes. This scaling technique enhances Efficient Net block performance by uniformly scaling the pre-processed image’s attributes, such as resolution, width, and depth, to determine the model's most appropriate attributes (Chen et al., 2023). Thereby, this block effectively retrieves features from pre-processed images, Pk and then it forwards the retrieved features to the STCF module for feature map generation.

Swin Transformer and CNN fusion module

After retrieving features from preprocessed images, Pk via IST and Efficient Net blocks in Transformer and CNN branches, respectively, from the encoder block. In the STCF module, spatial attention utilizes spatial relations between various feature maps. Consequently, these retrieved features are fused, but a specific module enhances informative features before the feature fusion process. Here, the used module is named the scSE module, which effectively enhances informative features by suppressing non-informative features with the help of two attention mechanisms (spatial Squeeze & Excitation (sSE) branch and channel Squeeze & Excitation (cSE) branch) (Chen et al., 2023). Notably, this spatial and channel Squeeze & Excitation (scSE) module works by applying spatial attention to attain Tr1xandCNN1x features utilizing features retrieved from the Transformer branch, Trx and CNN branch, CNNx, respectively. After enhancing informative features retrieved by Transformer and CNN branches, this module fuses these features and feeds these fused features into the scSE module for recalibrating it as Tr^xandCNN^x, respectively. The overall functioning of the STCF module is mathematically formulated from Eqs. (7)–(10) (Chen et al., 2023). In Eq. (7), Tr1xandCNN1x are denoted as input features from the CNN andTransformerbranchesrespectively.The fused features f^x are obtained by the following method for x = 0, 1, 2, 3:

(7) Tr1x=SA(Trx)CNN1x=SA(CNNx)

(8) Tr^x=scSE([CNN1x,Trx])CNN^x=scSE([Tr1x,CNNx])

(9) b^x=conv(Trxw1x⊙CNNxw2x)

(10) f^x=re([Tr^x,CNN^x,b^x])

where conv is a 3 × 3 convolution, w1xandw2x denotes the weights of feature maps retrieved from Transformer and CNN branches and are numerically valued by real numbers. In Eq. (9), ⊙ represents Hadamard product. Tr^xandCNN^x are the recalibrated features from the CNN. Therefore, this module’s fused feature maps resulted after sending to the Decoder block and Skip connection. In which segmentation takes place, and the resultant segmented image from the improved Transformer-based CNN model is symbolized as Sk.

The improved Transformer-based CNN (ITCNN) model offers several key advantages for segmentation tasks, particularly in medical imaging. Its dual-branch architecture, combining the improved Swin Transformer for contextual understanding with EfficientNet for detailed spatial feature extraction, ensures a comprehensive and robust feature representation. The Swin Transformer and CNN Fusion Module (STCF) efficiently merge these features, enhancing segmentation accuracy by integrating global context with fine-grained spatial details. The model’s use of a decoder block and skip connections further refines and upscales the segmented output, preserving high resolution and precise localization of features.

Feature extraction in terms of texture, shape, color & deep features

Feature extraction is a crucial step in processing an image. It involves transforming the image into features that can be analyzed while preserving the original data. Once the image is segmented, informative features are extracted. The most informative features that can be extracted from segmented images Sk Include texture features, shape features, color features, and deep features.

Texture features

Generally, the features of texture define the various characteristics or attributes extracted from an image, which specifically indicate the spatial patterns and structures present within the segmented image, Sk.The significant role of extracting texture features from segmented images denotes its informative data and defines and differentiates the diseased and non-diseased regions within it. The two types of techniques employed in the extraction of texture features are, 1) Extraction of Texture features using modified LGIP

2) Extraction of Texture features using multi-texton.

1. Modified LGIP-based texture feature extraction

Generally, LGIP is a familiar local texture feature-extracting technique that explicitly captures information about the local gradients within an image. Likewise, Local Binary Pattern (LBP) is another local texture feature extraction method that captures an image’s local structure by comparing its pixel intensity values in a neighbourhood around each centre pixel. These two local texture feature-extracting techniques are mainly used individually in existing research. These two techniques are combined and formed as a modified LGIP. The procedural process involved in the modified LGIP technique using improved LBP is shown in the sections below.

(i) Improved LBP in modified LGIP technique

The conventional LBP local texture feature extracting technique will allocate each pixel in an image with eight binary bits based on its intensity value with eight neighbours. Then, the resultant output from conventional LBP is transformed into binary format (Wady, 2022). This transformed binary representation from conventional LBP is mathematically formulated in Eq. (11), where ip is the intensity of pixel, p in a neighbourhood around the central pixel, and the pixel intensity of the centre pixel is indicated as ic. Here, the segmented image, Sk is subjected to the extraction of improved LBP using the formulation Eq. (12). Initially, the segmented image, Sk is transformed into a grey-scale image, gSk. In the grey-scale image, the grey value of the eight nearby pixels in a neighborhood around the central pixel is indicated, and its value is evaluated in an improved manner formulated in Eq. (13).

(11) LBP=∑i=0n−1⁡2ith(gi−gc)where,th(Sk)={1,Sk≥00,Sk<0

(12) ILBP=∑i=07⁡2ith(gi−gc)where,th(Sk)={1,Sk≥00,Sk<0

(13) gi=(pi2−1|pi|adj(pi))(Lpi+RpiTpi)

where, pi denotes one of the eight nearby pixels in a neighborhood around the central pixel and Lpi,RpiandTpi indicates the left, right, and top pixels of pi. Thus, from this improved LBP technique, the spatial patterns in segmented image, Sk are captured effectively by following the above procedure.

(ii) Process executed after the onset of improved LBP Technique in modified LGIP

After improved LBP procedure, the procedure of general LGIP follows. Generally, LGIP partitions an image through a fractional partitioning technique for generating binary vectors corresponding to the image’s horizontal and vertical directions. This modified LGIP technique results from texture features effectively by providing information about the local gradient variations based on the distribution and arrangement of gradients within the defined local regions or neighborhoods in segmented images, Sk. In this modified LGIP, eight nearby pixels considered in improved LBP are initially utilized to compute the response of gradients in each pixel’s eight potential directions by applying the Sobel gradient operator.

In this instance, gradient responses in eight orientations at each pixel are calculated using Sobel masks (M0 to M7). Then, using the signs of the eight replies, an eight-bit code is allocated to the pixel. The modified local gradient pattern (LGIP) technique uses an extension of the local binary pattern (LBP) method to capture more detailed texture information by considering the gradient responses in multiple orientations. The Sobel operator is typically used to calculate gradient responses in the horizontal and vertical directions, but for the modified LGIP, it seems that the Sobel operator has been extended to calculate gradients in eight different orientations. The parameters M0 to M7 likely represent the masks or kernels for the Sobel operator in these eight different orientations. In the standard Sobel operator, there are two 3 × 3 kernels: one for detecting horizontal changes and one for vertical changes. For the modified LGIP, these kernels have been adapted or extended to capture gradient information in eight specific directions around a pixel. For the modified LGIP, the kernels M0 to M7 would be similar in concept but oriented in different directions to capture gradients at angles other than just horizontal and vertical. These could include diagonal and intermediate orientations.

Each pixel in an image is assigned an eight-bit value based on the polarity of its gradient value. The encoding process involves setting a single bit to 1(p) and the rest to 0, resulting in each bit representing the output of a specific mask.

The eight bits can be evaluated using intensity comparisons between the central pixel and its neighbors to speed up computations.

Improved LBP uses the Sobel gradient operator to stabilize illumination variation and noise. It is faster than LDP’s Kirsch edge detector and achieves gradient operator from the central pixel to neighbours in stable conditions (Peng, Zhang & Guo, 2023). Thus, the modified LGIP captures local texture features effectively by following these procedures.

2. Multi-texton Analysis-based Texture Feature Extraction

Multi-texton analysis is done by extracting and analyzing diverse textons in an image. Textons are the primitive patterns or fundamental elements in an image that constitute the texture of it. Initially, this multi-texton analysis technique works on extracting diverse sets of textons from the segmented image, Sk for capturing several textons representing different texture patterns. Then, these extracted textons are encoded into an appropriate feature representation. Then, these features are mapped to a feature space to preserve their distinctive properties (Wady, 2022). Thus, it results in informative local texture features effectively from the segmented image, Sk. After the extraction of the local texture features using both the modified LGIP and Multi-texton techniques, they are combined and symbolized as TexF. The use of LGIP enhances texture feature extraction by capturing finer gradient variations, while multi-texton analysis provides a richer representation of texture patterns. This dual approach offers a more comprehensive understanding of the image features, which is crucial for accurate disease classification. The modified local gradient increasing pattern (LGIP) and Multi-texton analysis offer significant advantages in feature extraction by enhancing texture sensitivity and feature discrimination. LGIP improves texture detail capture by focusing on local gradient variations, which aids in distinguishing subtle patterns and structures in medical images while maintaining robustness against noise. Multi-texton analysis complements this by providing a comprehensive representation of diverse texture patterns, enriching the feature set and enhancing the model’s ability to recognize and classify complex textures. Together, these methods improve segmentation and classification accuracy, making them highly effective for analyzing intricate and varied imaging data.

Shape features

Shape features are quantitative measures or descriptors that characterize an image’s geometric properties and objects or regions. These features provide valuable information about objects’ contours, boundaries, and structural attributes, allowing tasks like differentiation and classifying objects based on their shapes. The common mechanisms used for shape feature retrieval include identification such as aspect ratio, lines, circularity, and boundaries and detecting areas of variation or stability by region growth and edge detection approaches (Minarno et al., 2016). Thereby, from the segmented image Sk, the obtained shape features are symbolized as ShaF.

Color features

Color features are quantitative measures that describe the characteristics of colors present in an image, mainly used in applications like image retrieval, content-based image analysis, object recognition, etc. The color feature provides important details on the arrangement, makeup, and characteristics of colors in an image. Based on color similarity, color features are extracted from photos by calculating a color histogram for each image, which identifies the percentage of pixels in an image that contains a certain value. The color feature will segment color proportion by region and spatial relationship among several color regions (Minarno et al., 2016). The input segmented image is applied to extract such practical features in this feature extraction step, and the resultant color feature is symbolized as ColF.

Deep features

Deep features from an image are extracted using DL models, training them using high-level image patterns. These features are derived from layers deep within the DL techniques’ network architecture from which more complex features are captured. The captured complex features give more meaningful information about the input data than the raw input. In this proposed work, deep features from segmented images, Sk is extracting by employing two efficient DL architectures, such as ResNet and VGG16.

1. ResNet-based Deep Feature Extraction

ResNet (Harini & Lalitha Bhaskari, 2011) is an invention of Microsoft Corporation that became popular in image classification. With residuals, this architecture is improved because an input image is converted to an output without the need for any neural network operations, preserving the original image. A deep residual network that traverses several levels was used to create Resnet. Initially, this ResNet architecture is trained on a lung medical image dataset to prepare it as a pre-trained ResNet architecture with weights for efficiently capturing deep and informative features from the segmented image, Sk.

2. VGG16-based Deep Feature Extraction

Generally, VGG16 architecture (Sriporn et al., 2020) consists of 16 layers, where 13 are convolutional, and three are fully connected. These layers are involved in extracting more complex features from segmented images, Sk by enabling VGG16 to capture the visual content at various levels of abstraction. Deep features extracted by VGG16 architecture from segmented images, Sk are obtained from the output of each convolutional layer as the input segmented image, Sk p passes through the architecture; each convolutional layer extracts high-level representations of cap S to the k. Further, the deeper layers in this architecture capture more complex patterns and structures in the input-segmented image, such as textures, shapes, and object components. Thus, this architecture extracts more informative and deep features efficiently.

After the extraction of deep features using both the ResNet and VGG16 architectures, they are combined and symbolized as DeF.

Finally, the features extracted from the segmented image, Sk Using specific techniques are concatenated, and it is symbolized as Fk: Fk=[TexFShaFColFDeF].

Classification by hybrid L-MLSTM model

After the above steps, lung disease classification is done by employing a novel hybrid model, L-MLSTM, formed by integrating LinkNet and MLSTM models. The architecture of the proposed L-MLSTM model is illustrated in Fig. 3. LinkNet is one of the DL models, the upgraded version of the U-Net architecture, incorporating skip connections to enable accurate and reliable classification. LinkNet’s architecture includes an encoder-decoder design that excels at semantic segmentation by progressively refining features through a series of convolutions and upsampling operations. This is crucial for accurately segmenting affected regions in lung images. The use of skip connections helps preserve spatial information, which is essential for precise segmentation tasks where fine details are critical. LinkNet’s design allows for high-resolution output, which is important for detailed segmentation of lung structures and abnormalities. However, the LinkNet model (Poornima & Pushpalatha, 2019) may not be more efficient due to its limitation of overfitting, which occurs when the model is too specialized in segmenting the specific features in the training data. Subsequently, the conventional LSTM (Briskline Kiruba, Petchiammal & Murugan, 2022) model’s performance may be affected by overfitting issues when it is keen to memorize excess features in the training data. This eventually reduces its generalizability to unseen data. So, a modified LSTM model, i.e., MLSTM, is implemented more efficiently using a new activation function for efficient lung disease classification. Modified LSTM adds the ability to process and learn from sequential data, providing a deeper understanding of the disease’s temporal dynamics. The introduction of the ScReLU activation function combines multiple activation functions (Sigmoid, comb-H-sine, and Leaky ReLU), aiming to offer a more flexible and powerful non-linearity. This can help in capturing complex patterns and improving the model’s learning capability. By using ScReLU, the modified LSTM can potentially achieve better performance in terms of learning efficiency and model accuracy, which is crucial for medical image analysis where precision is key. Further, the LinkNet and MLSTM models are integrated to form a hybrid model for lung disease classification to obtain efficient classification results. This integrated model addresses both spatial and temporal dimensions, providing a comprehensive tool for medical image analysis and contributing to advancements in diagnostic accuracy and patient care.

Figure 3 Proposed L-MLSTM model’s architecture for classifying lung disease.

LinkNet in L-MLSTM model

Based on pixel-wise classification, the LinkNet model effectively computes and outputs lung disease classification results. The encoder and decoder routes have several encoder and decoder blocks, respectively.

The informative features, Fk extracted from the segmented image, are given as input to each encoder block comprising convolutional layers, batch normalization, and ReLU activation functions. Samples of those input features are blocked by the encoder to obtain the local and global context information. The decoder blocks in the decoder path use feature-up sampling to provide the original image resolution, in contrast to encoder blocks. Therefore, to enable relevant data propagation at varied scales and orientations, each decoder block uses skip connections to mix features from the encoder block. This allows for the progressive recovery of the spatial data lost during the encoder path's down-sampling procedure. Here, the involvement of skip connection is essential to efficiently transfer low-level and high-level features by signifying the precise localization of objects, boundaries, and contours in the segmented image Sk Finally, the architecture of this model is concluded with a final convolutional layer that passes generated classification results on lung diseases ( LinkNC) through a softmax activation function.

MLSTM in L-MLSTM model

This MLSTM is developed as the conventional LSTM model, but the difference is that this implemented MLSTM used a new activation function named ‘ScReLU’ instead of a sigmoid activation function. This replacement is introduced for enhancing the non-linearities introduced into the model to train and approximate intricate, non-linear relations within the segmented image; along with this ScReLU activation function, the MLSTM model is implemented based on conventional LSTM architecture, which means that it is implemented using cell state ct, input gate it, output gate ot, candidate vector ct′, hidden state ht, and forget gate ft using the following equations. The MLSTM model functions initially by deciding which new information should be stored in the cell state where the input gate makes this decision, it in it. Then, the input gate generates an input vector by considering the previous hidden state and current input, merged with the candidate vector, ct′ for updating the cell state; the cell state acts as a memory of the MLSTM model by allowing functions such as storing and accessing information. The forget gate can determine what data from the previous cell state should be kept or discarded. Further, it initializes the inputs as current input and previous cell state and a forget vector as output; it is then formulated under element-wise multiplication with the last state cell for removing unnecessary data. Finally, the classification results ( MLSTMC) on lung disease classification are regulated by the output gate by controlling which components of the cell state are used in the computation of the hidden state ( ht) output. Iteration continues by feeding the recurrent part of the input vector after updating this part using the current cell state (an input vector comprises two components: Input and recurrent parts). The updation of each gate and cell state is mathematically formulated from Eqs. (14) to (19) (Poornima & Pushpalatha, 2019).

(14) it=ScReLU(wi[Fk(t),ht−1]+bii)

(15) ft=ScReLU(wf[Fk(t),ht−1]+bif)

(16) ot=ScReLU(wo[Fk(t),ht−1]+bio)

(17) ct′=ScReLU(wc[Fk(t),ht−1]+bic)

(18) ct=ft∗ct−1+it∗ct′

(19) ht=ot∗ScReLU(ct)

where the Fk(t) is the input vector, ht−1 represents the previous state hidden vector, weights and bias vectors of all gates and cell state are represented as W and b. The employed activation function Sigmo-Comb Rectified Linear Unit (ScReLU), is obtained by combining three activation functions: Sigmoid(f(Fk)=1(1+exp−Fk)), comb-H-sine (f(Fk)=sinh⁡(βFk)+1sinh(βFk)), and Leaky ReLU (f(Fk)={Fk;ifFk≥0αFk;otherwise) (Vijayaprabakaran & Sathiyamurthy, 2022). To express the formulation of the ScReLU activation function, Sigmoid (Xiangyang et al., 2023) and comb-H-sine (Guo et al., 2021) activation functions are added, and the result from the addition operation is shown in Eq. (20). Combining a bounded function (like Sigmoid) with an unbounded function (like Comb-H-Sine) provides a powerful activation function that can normalize outputs while retaining the ability to handle extreme values and capture complex patterns with the flexibility of unbounded functions. It improve gradient flow and learning efficiency across a wide range of input values and enhance the model’s regularization and generalization capabilities. This ensures that the activation function can provide both stability and flexibility, addressing potential limitations of using each function individually.

(20) f(Fk)=(1+exp−Fk)−1+sinh⁡(βFk)+1sinh(βFk)

(21) LeakyReLU⇒f(Fk)={Fk;ifFk≥0αFk;otherwise

Now, Eqs. (20) and (21) are merged to form the ScReLU activation function, which is formulated in Eq. (22), where α is equivalent to 0.01. The ScReLU activation function combines the strengths of multiple activation functions; Sigmoid, comb-H-sine, and Leaky ReLU to provide a more versatile and effective non-linearity for neural networks. This composite function enhances the model's ability to learn complex patterns by improving gradient flow and mitigating issues like vanishing gradients through its smooth transitions and robust handling of negative inputs. ScReLU offers greater stability and adaptability across various input ranges, making it particularly useful for tasks requiring nuanced feature extraction and improved convergence. Its ability to blend different activation behaviors results in richer feature representations and faster, more stable training, thereby enhancing overall model performance. ScReLU’s combination of Sigmoid, Comb-H-Sine, and Leaky ReLU functions offers a flexible, adaptive approach that can handle a broad range of input values and non-linearities effectively. This makes it particularly suitable for complex tasks like lung disease segmentation and classification, where diverse and varying features need to be captured and processed. Its ability to manage gradient flow and avoid saturation effects further enhances its utility in deep learning models for medical image analysis.

(22) ScReLU⇒f(Fk)={Fk;ifFk≥0(1+exp−Fk)−1+sinh⁡(βFk)+1sinh(βFk);if0>Fk≥−1;(−∞,∞)αFk;otherwise

where, β is a parameter for the comb-H-sine function, α is the slope for Leaky ReLU.

For Fk≥0: The ScReLU function returns the input value Fk similar to the ReLU function.

For −1≤Fk<0: The function combines a sigmoid component with the comb-H-sine function, which captures a range of non-linear behaviors for slightly negative values. The (−∞,∞) ranges with both positive and negative values, thus maintain meaningful gradients and handle extreme input values effectively, ensuring robust learning and performance across a wide range of input values.

For Fk≤−1: The function uses Leaky ReLU to ensure the gradients are not zero for highly negative values, preventing issues with gradient flow during training.

ScReLU stands out as a robust activation function due to its ability to introduce complex non-linearities through the combination of multiple activation functions. It improve gradient flow and mitigate issues like vanishing gradients and inactive neurons. It handle extreme values effectively, ensuring that the activation function remains effective across a wide input range. It provide adaptive behavior, allowing the model to learn efficiently from diverse input values. These properties make ScReLU a compelling choice for enhancing model performance, particularly in complex and varied tasks.

Figure 4 displays the activation function plot of the ScReLU function over traditional activation functions. The ScReLU activation function offers better efficiency over traditional activation functions like Sigmoid, comb-H-sine, and Leaky ReLU. ScReLU remains at zero for negative inputs and linearly increases with a positive slope for positive inputs, while other activation functions such as; the sigmoid function start near zero, slowly increase, and then plateaus at an output of 1 as input increases. The comb-H-sine function oscillates above and below zero without a clear trend as input increases. Leaky ReLU has a small, positive slope for negative inputs and a larger positive slope for positive inputs. Thus it is evident that the ScReLU activation function shows better performance. ScReLU avoids the vanishing gradient problem and is computationally more efficient, making it a better choice for deeper networks. ScReLU provides more stable and predictable outputs compared to the oscillatory nature of comb-H-sine. The scaling factor can be adjusted to improve the learning process, making it more flexible than standard ReLU. The use of ScReLU in Modified LSTM networks further enhances the model’s performance by providing stable and efficient learning for lung disease segmentation and classification.

Figure 4 Activation function plot: proposed ScReLU over traditional activation functions.

The hybrid model that combines LinkNet architecture with modified long short-term memory (LSTM) networks offers substantial advantages for lung disease classification by leveraging the strengths of both components. LinkNet’s encoder-decoder structure provides precise spatial feature extraction and segmentation through its efficient skip connections, ensuring detailed and accurate identification of lung abnormalities. Meanwhile, the modified LSTM network, enhanced with the ScReLU activation function, captures complex temporal dependencies and contextual information, improving the model's ability to understand sequential patterns and variations over time. This integration of spatial detail and sequential learning results in superior classification performance, allowing the model to effectively differentiate between various lung diseases and enhance diagnostic accuracy. Finally, the classification results on lung diseases using LinkNet and MLSTM models are averaged for efficient and accurate results on lung disease classification. The final results are symbolized as Ck. Table 2 describes the parameters of the classifiers.

Table 2 Parameters of the classifiers.

Models	Parameter values	
Bi-GRU	Bidirectional layer1-GRU layer Units-124	
Bidirectional layer 2-GRU layer Unit-64	
Dropout layer-Rate-0.5	
Dropout layer-Rate-0.2	
Dense layer	
Activation-softmax	
Los-categorical cross-entropy	
Optimizer-adam	
Metrics-accuracy	
Epoch-10	
Batch size-100	
LSTM	LSTM layer Units-128	
Dense layer	
Activation-softmax	
Loss-sparse categorical cross entropy	
Optimizer-adam	
Metrics-accuracy	
Epoch-10	
Batch size-100	
CNN	Convolution layer	
1 Dimensional	
Kernel size-1	
Filters-64	
Padding-valid	
Activation-relu	
Dropout layer	
Rate-0.5	
Maxpooling layer	
1 dimensional	
Pool size-1	
Flatten layer	
Dense layer	
Activation-softmax	
Loss-sparse categorical cross entropy	
Optimizer-rmsprop	
Metrics-accuracy	
Epoch-10	
Batch size-100	
RNN	Simple RNN layer	
Units-64	
Activation-relu	
Dense layer	
Activation-softmax	
Loss-sparse categorical cross entropy	
Optimizer-adam	
Metrics-accuracy	
Epoch-10	
Batch size-100	
VGG19	Optimizer-adam	
Loss-sparse categorical cross entropy	
Metrics-accuracy	
Epoch-10	
Modified LSTM	LSTM layer Units-128	
Dense layer	
Activation-modified activation (ScReLU)	
Loss-sparse categorical cross entropy	
Optimizer-adam	
Metrics-accuracy	
Epochs-25	
Batch size-50	
LinkNet	Input layer	
Encoder block 1	
Convolutional layer	
2 Dimensional	
Filter size-32	
Kernel size-(3, 3)	
Padding-same	
Batch normalization layer	
ReLU layer	
Convolutional layer	
2 Dimensional	
Filter size-32	
Kernel size-(3, 3)	
Padding-same	
Batch normalization layer	
ReLU layer	
Max pooling layer	
2 Dimensional	
Pool size-(2, 2)	
Encoder block 2-filter size-64	
Encoder block 3-filter size-128	
Encoder block 4-filter size-256	
Global average pooling layer	
2 Dimensional	
Dense layer	
Activation-softmax	
Loss-categorical cross entropy	
Optimizer-adam	
Metrics-accuracy	
Epochs-25	
Batch size-64	

Result and discussion

Experimental setup

Python 3.7.9 is used to implement the proposed L-MLSTM model. By contrasting traditional methods, including HDE-NN (Gugulothu & Balaji, 2024), DBN (Manickavasagam & Sugumaran, 2023), LSTM, LINKNET, SVM, Bi-GRU, RNN, CNN (Demir, Sengur & Bajaj, 2020), and VGG19 + CNN (Alshmrani et al., 2023), the L-MLSTM model is effectively validated. COVID-19 normal pneumonia-CT images (https://www.kaggle.com/datasets/anaselmasry/covid19normalpneumonia-ct-images) and Chest CT scan images (https://www.kaggle.com/datasets/mohamedhanyyy/chest-ctscan-images) are the datasets used for training and testing lung disease detection. Table 3 describes the system configuration.

Table 3 System configuration.

Device specifications	Device specifications	
Processor	AMD Ryzen 5 3450U with Radeon Vega Mobile Gfx 2.10 GHz	
Installed RAM	16.0 GB (13.9 GB usable)	
System type	64-bit operating system, x64-based processor	
Windows specifications	
Edition	Windows 11 Home Single Language	
Version	23H2	

Dataset description

Dataset 1

One of the datasets utilized for L-MLSTM model training and testing is COVID-19 normal pneumonia-CT images. The dataset has CT images of normal, pneumonia, and COVID-19 patients’ lungs. In the Covid2CT category, there are 2,034 images; in the normal lungs category, it provides 2,119 images; the pneumonia category folder contains 3,390 images. Each image is 512 × 512 pixels.

Dataset 2

The second dataset utilized for model testing and training is pictures from chest CT scans. The material covers three forms of chest cancer: adenocarcinoma, large-cell carcinoma, and squamous-cell carcinoma. The total number of data presented is 1,000. The pictures are in JPG or PNG format. The main folder contains step folders comprising a 70% training set, a 20% testing set, and a 10% validation set. CT scan photos from the dataset show several forms of chest cancer. Table 4 describes the training, testing, and validation data for both datasets.

Table 4 Training, testing, and validation data for datasets 1 and 2.

Learning percentage	Training	Testing	Validation	
Dataset 1	
60	630	420	420	
70	735	315	315	
80	840	210	210	
90	945	105	105	
Dataset 2	
60	600	400	400	
70	700	300	300	
80	800	200	200	
90	900	100	100	

Comparative study

To assess the effectiveness and efficiency of the L-MLSTM technique, an assessment with the L-MLSTM model and the traditional models for datasets 1 and 2 is made using a range of measures. A comparison is made between the L-MLSTM approach and traditional methods based on metrics like accuracy, sensitivity, specificity, precision, recall, F2-score, F-measure, and NPV. FPR and FNR are the measures used for error detection. The selected metrics provide a comprehensive evaluation of the model’s performance across different aspects, ensuring a balanced assessment of its effectiveness, especially in critical medical applications. They collectively offer insights into how well the model identifies both positive and negative cases, its reliability, and its overall diagnostic accuracy. Each of these metrics provies benefits in the comparative analysis, they are as follows:

Positive metrics: Accuracy measures the overall correctness of the model’s predictions. It provides a general sense of the model’s performance but can be misleading if the dataset is imbalanced. Sensitivity assesses the model’s ability to correctly identify positive cases. It is crucial for medical diagnostics where missing a positive case (e.g., a disease) can have serious implications. Specificity evaluates the model’s ability to correctly identify negative cases and it is important for ensuring that healthy individuals are accurately classified, minimizing false positives. Precision measures the accuracy of positive predictions. It’s vital when the cost of false positives is high, such as in confirming a diagnosis. Recall is a metric that measures how often a machine learning model correctly identifies positive instances (true positives) from all the actual positive samples.

Negative metrics: FPR indicates the rate at which negative instances are incorrectly classified as positive. It’s important for assessing the risk of false alarms. FNR measures the rate at which positive instances are incorrectly classified as negative. It is crucial for understanding the risk of missed diagnose.

Other metrics: F2-score balances precision and recall with more emphasis on recall. Useful in scenarios where missing a positive case (high recall) is more critical than the precision of positive predictions. F-measure combines precision and recall into a single metric, offering a balance between the two. It is helpful in evaluating the trade-off between precision and recall. NPV measures the proportion of true negatives among all negative predictions. Important for understanding the reliability of negative classifications.

In this study, we selected the hybrid L-MLSTM model due to its superior performance compared to other state-of-the-art models. This choice is based on empirical results showing that L-MLSTM excels in handling both spatial and temporal aspects of lung disease analysis. The state of art models such as HDE-NN (Gugulothu & Balaji, 2024), DBN (Manickavasagam & Sugumaran, 2023), CNN and VGG19 + CNN are included in the comparison. While CNNs excel in feature extraction from static images, they may struggle with capturing complex patterns over time or across multiple contexts. The addition of LSTM to our hybrid model enhances its ability to analyze sequential or temporal changes, which is a limitation in pure CNN architectures. While HDE-NN excels in parameter optimization, the L-MLSTM model offers a more holistic approach by integrating advanced segmentation techniques with sequential analysis. This results in potentially better handling of complex medical imaging tasks, where both feature extraction and temporal dependencies are crucial. DBNs focus on learning data distributions and features, but they may not handle sequential or temporal data as effectively as LSTMs. The L-MLSTM model’s ability to integrate sequential data processing with advanced feature extraction provides a significant advantage in tasks involving dynamic or multi-slice data. VGG19 + CNN models are strong in feature extraction but do not inherently handle sequential data or provide global context. The L-MLSTM model’s integration of Transformers and Modified LSTMs offers a more comprehensive solution by combining spatial, contextual, and sequential analysis. Also, the traditional classifiers includes LSTM, LINKNET, SVM, Bi-GRU, and RNN are used for the comparative analysis.

Preprocessing analysis

For datasets 1 and 2, the study of picture preprocessing using the mean filter and existing filters such as the Clahe, Gaussian, and mean filters is shown in Figs. 5 and 6. Looking at the results, the preprocessed image using the median filter has a more appealing appearance. The value of each pixel is replaced by the median of its surrounding pixels when using median filtering. The median reduces noise effectively without adding erroneous pixel values because it is less susceptible to extreme values, or outliers. For improved results, median filtering is a useful preprocessing method for lung pictures.

Figure 5 Preprocessing analysis on dataset 1 (A) input image (B) Clahe filter (C) Gaussian filter (D) Mean filter (E) Median filter.

Figure 6 Preprocessing analysis on dataset 2 (A) input image (b) Clahe filter (C) Gaussian filter (D) Mean filter (E) Median filter.

Table 5 presents the enhanced median filter’s performance for datasets 1 and 2 in terms of SSIM and PSNR in comparison to the current filters, which include Gaussian filtering, Clahe filtering, and conventional median filtering. Two computational tools that were commonly used in the evaluation of image quality were PSNR and SSIM. Based on the relationship between each pixel and the other pixels in an 11-by-11 neighborhood, the SSIM function determines the structural similarity index of each pixel. Table 5 makes clear that the improved median filter produced higher PSNR and SSIM rates—roughly 30.47382 and 0.903462 in Dataset 1 and 34.22451 and 0.914578 in Dataset 2, in that order. In contrast, gaussian, low pass, and conventional median filters obtained subpar results. As a result, compared to previous filters, the enhanced median filter ensures a more effective segmentation procedure.

Table 5 Analysis of preprocessing methods in terms of PSNR and SSIM.

	PSNR	SSIM	
Dataset 1	Dataset 1	Dataset 1	
Clahe	22.7528	0.803589	
Mean	21.13795	0.715685	
Gaussian	24.23645	0.851893	
Median	30.47382	0.903462	
Dataset 2	Dataset 2	Dataset 2	
Clahe	16.18248	0.619834	
Mean	25.43892	0.768613	
Gaussian	28.62405	0.882414	
Median	34.22451	0.914578	

Segmentation analysis

For datasets 1 and 2, the picture results of the pre-processed input image, TCNN, Segnet image, U-Net, fuzzy C-mean clustering, K-mean clustering, and ITCNN technique are represented in Figs. 7 and 8. An analysis is done on the segmented images by the ITCNN approach and conventional models. The ITCNN model for segmentation uses an Improved transformer-based CNN model, which enhances segmentation phase performance. The ITCNN model's segmentation accuracy is examined and contrasted with methods from the past. Some of the segmentation evaluation metrics are the dice score and the Jaccard similarity. The segmentation assessment of the recommended approach in comparison to traditional methods is displayed in Tables 6 and 7.

Figure 7 Segmentation outcome of dataset 1 (A) preprocessed image, (B) TCNN (C) Fuzzy C-mean Clustering, (D) K-mean clustering, (E) SEGNET image, (F) U-Net, (G) ITCNN.

Figure 8 Segmentation outcome of dataset 2 (A) preprocessed image, (B) TCNN, (C) fuzzy C-mean clustering, (D) K-mean clustering, (E) SEGNET image, (F) U-Net, (G) ITCNN.

Table 6 Segmentation analysis for dataset 1.

Measure	FCM	k-means	U-NET	SEGNET	Conventional	ITCNN	
DICE	0.6480	0.6352	0.7511	0.6752	0.7185	0.8497	
JACCARD	0.6325	0.6248	0.8396	0.7244	0.8268	0.9485	
Segmentation accuracy	0.5880	0.6283	0.8495	0.7520	0.8331	0.9539	

Table 7 Segmentation analysis for dataset 2.

Measure	FCM	k-means	U-NET	SEGNET	Conventional	ITCNN	
DICE	0.6609	0.6606	0.7561	0.6826	0.7186	0.8642	
JACCARD	0.6501	0.6231	0.8428	0.7263	0.8318	0.9449	
Segmentation accuracy	0.6488	0.6693	0.8345	0.7436	0.8228	0.9550	

Dice score: A measure of how similar two samples are, the Dice score (sometimes called the Dice coefficient) is computed by dividing the size of their intersection twice by the total of the two samples’ sizes.

(23) Dicescore=(2|A∩B|)/(|A|+|B|)

Jaccard similarity: A metric compares sample sets’ variety and similarity. It can be expressed as the product of the size of the sets’ union and the size of their intersection.

(24) J(A,B)=(∣A∩B∣)/(∣A∪B∣)

Analysis of confusion matrix

The efficacy of the L-MLSTM model in comparison to conventional techniques is assessed using a confusion matrix by examining true positive, true negative, false positive, and false negative results. Figures 9 and 10 display the L-MLSTM Confusion Matrix and traditional methods for datasets 1 and 2. Higher TPR means fewer missed cases of lung disease, which is critical for early diagnosis and treatment. Higher TNR reduces the incidence of false positives, ensuring that healthy individuals are not wrongly identified as diseased. The L-MLSTM model achieved a true positive rate (TPR) of 265 and a true negative rate (TNR) of 579, which are the highest among all conventional models. This indicates that the L-MLSTM model is exceptionally effective at correctly identifying both positive (diseased) and negative (healthy) cases. In comparison, conventional methods have lower TPR and TNR values, with the highest TPR being 228 and TNR being 542. The higher TPR signifies that the L-MLSTM model excels in detecting the presence of lung disease, reducing the risk of missed diagnoses. The higher TNR implies it is also better at correctly identifying non-diseased cases, thus reducing false alarms.

Figure 9 Confusion matrix of dataset 1 (A) CNN (B) VGG19 + CNN (C) Bi-GRU (D) Link-Net (E) LSTM (F) RNN (G) SVM (H) HDE-NN (I) DBN (J) L-MLSTM.

Figure 10 Confusion matrix of dataset 2 (A) CNN (B) VGG19 + CNN (C) Bi-GRU (D) Link-Net (E) LSTM (F) RNN (G) SVM (H) HDE-NN (I) DBN (J) L-MLSTM.

Similarly, the L-MLSTM model outperforms conventional approaches with a TPR of 264 and a TNR of 863. These results suggest that the model consistently performs better in both detecting and avoiding false positives, making it highly reliable for lung disease detection across different datasets. The model’s superior TPR and TNR contribute to overall reliability in clinical settings, where accurate diagnosis is crucial for patient management. Subsequently, lower FNR reduces the risk of overlooking true cases of lung disease, leading to better patient outcomes. Lower FPR helps in reducing unnecessary follow-up tests and psychological stress for patients falsely identified as having a disease. Fewer false positives and negatives improve the overall efficiency of the diagnostic process, saving time and resources in medical settings. The L-MLSTM model has a false negative rate (FNR) of 50 and a false positive rate (FPR) of 51, both of which are lower compared to conventional techniques. Lower FNR means fewer false negatives, thus reducing the risk of failing to detect actual cases of lung disease. A lower FPR means fewer false positives, which minimizes unnecessary concern and further testing for healthy individuals. The FNR is even lower at 36, which further enhances the model’s ability to correctly identify diseased cases without missing any. The reduced FNR in dataset 2, along with high TPR and TNR, suggests that the model is robust and consistently performs well across different datasets.

The results indicate that the L-MLSTM model not only performs better than conventional methods but also shows robustness across different datasets. The combination of LinkNet and modified LSTM allows the model to capture both spatial and temporal features more effectively, leading to better segmentation and classification performance. The use of advanced feature extraction techniques, such as modified LGIP and Multi-texton analysis, contributes to improved accuracy by providing more detailed and relevant information for classification. Thus, the L-MLSTM model demonstrates significant advantages over conventional techniques, with higher TPR and TNR, and lower FNR and FPR. These results highlight the model's effectiveness in accurately diagnosing lung diseases while minimizing false results.

Interpretation of both positive and negative metrics

The L-MLSTM model is evaluated over several previously developed methods such as HDE-NN (Gugulothu & Balaji, 2024), DBN (Manickavasagam & Sugumaran, 2023), LSTM, LINKNET, SVM, Bi-GRU, RNN, CNN, and VGG19 + CNN, for lung disease detection for datasets 1 and 2. The comparison is made at 60, 70, 80, and 90 learning percentages. As demonstrated in Fig. 11A, for dataset 1, the L-MLSTM approach's superior accuracy level of lung disease detection at a learning percentage of 70 is 89%, which was greater than the conventional models. Furthermore, at learning percentages of 80 and 90, the L-MLSTM model achieved sensitivity of 87% and 92%, respectively. The specificity, precision, and recall for the suggested work were 96%, 93%, and 95% respectively, with a learning percentage of 90. Figure 12 displays the performance assessment of the L-MLSTM model evaluated over conventional approaches for dataset 2. At an LP of 90, the L-MLSTM model’s accuracy was 95%, nearly 8% greater than previous models. The sensitivity and Precision of the L-MLSTM model are 88% for both, which is higher than the conventional technique. LinkNet and modified LSTM are used to classify lung disease. High accuracy signifies that the L-MLSTM model correctly identifies both diseased and healthy cases with greater precision. This is crucial for reliable diagnosis and effective treatment planning. Higher sensitivity reduces the risk of missed diagnoses, while increased specificity ensures fewer false positives. This dual improvement enhances the model’s reliability in distinguishing between affected and non-affected cases. High precision means that when the model predicts a disease, it is more likely to be correct, reducing unnecessary follow-ups and patient anxiety. As a result, the L-MLSTM can precisely detect lung disease. The L-MLSTM’s superior accuracy and sensitivity at higher learning percentages reflect its ability to learn and generalize better from training data. Improved specificity and precision indicate the model’s effectiveness in minimizing both false positives and negatives, leading to more reliable diagnoses.

Figure 11 Evaluation of the L-MLSTM model’s performance over conventional models on dataset 1 regarding (A) accuracy (B) sensitivity (C) precision (D) specificity, and (E) recall.

Figure 12 Evaluation of the L-MLSTM model’s performance over conventional models on dataset 2 regarding (A) accuracy (B) sensitivity (C) precision (D) specificity, and (E) recall.

A few negative metrics must be taken into account when evaluating a model’s performance. When comparing the model’s reduced error to other traditional techniques such as HDE-NN (Gugulothu & Balaji, 2024), DBN (Manickavasagam & Sugumaran, 2023), LSTM, LINKNET, SVM, Bi-GRU, RNN, CNN, and VGG19 + CNN for datasets 1 and 2, FPR and FNR values are taken into account. Figure 13 shows that the L-MLSTM model has the lowest FPR and FNR value rate for dataset 1. At the 80th learning percentage, the value of the FPR is approximately 0.03 as shown in Fig. 13A; the value of the L-MLSTM model was 5% smaller than that of the other conventional approaches. The FPR value of the L-MLSTM model, with a learning percentage of 90, is 0.07; however, the values of the other previous strategies were greater. Reduction of false positive lung illness diagnosis is associated with a lower FPR score. The FNR of the L-MLSTM approach is the lowest when compared to conventional procedures, with values of 0.12 at a learning percentage of 80 and 0.07 at a learning percentage of 90. The L-MLSTM approach’s negative metrics analysis for dataset 2 is assessed in Fig. 14. At the 90th LP, the L-MLSTM model has 0.02 FPR and 0.08 FNR; it gains comparatively low error value. Because the error measures in the suggested model have relatively low values, it performs better overall. Lower FPR means fewer healthy individuals are incorrectly identified as diseased, reducing unnecessary treatments and anxiety. Lower FNR ensures that fewer actual cases of disease are missed, leading to better patient outcomes. Lower FPR and FNR enhance the model’s reliability in clinical settings, improving trust in the diagnostic process and potentially leading to better patient care. The L-MLSTM model's reduced FPR and FNR are attributed to its hybrid architecture, which integrates advanced segmentation and classification techniques. This integration improves the model’s ability to differentiate between pathological and non-pathological conditions more effectively.

Figure 13 Analysis of the L-MLSTM approach in comparison to extant schemes for (A) FNR and (B) FPR for dataset 1.

Figure 14 Analysis of the L-MLSTM approach in comparison to extant schemes for (A) FNR and (B) FPR for dataset 2.

Figures 15 and 16 show the performance analysis of the proposed hybrid L-MLSTM model over conventional classifiers like HDE-NN (Gugulothu & Balaji, 2024), DBN (Manickavasagam & Sugumaran, 2023), LSTM, LINKNET, SVM, Bi-GRU, RNN, CNN, and VGG19 + CNN with regards to other measures such as F2-score, NPV, and F-measure for Dataset 1 and 2. The suggested model acquired better rates in NPV and F-measures for datasets 1 and 2. High F2-score, NPV, and F-measure values reflect the model’s ability to balance sensitivity and precision, providing a comprehensive assessment of its performance. These metrics ensure that the model performs well across various aspects of classification, including predicting negative cases correctly (NPV) and balancing precision and recall (F-measure). The suggested approach can detect and classify lung diseases precisely. The F2-score, NPV, and F-measure values of the L-MLSTM model highlight its capability to offer a well-rounded performance across different evaluation criteria. The hybrid nature of the model, combining LinkNet and Modified LSTM, likely contributes to its balanced and high-performance metrics. Thus the L-MLSTM model’s ability to achieve superior results in accuracy, sensitivity, precision, recall, and specificity, alongside lower FPR and FNR, and enhanced F2-score, NPV, and F-measure, demonstrates its effectiveness and reliability in lung disease detection. These results suggest that the L-MLSTM model offers a significant improvement over conventional technique. Its advanced hybrid architecture, which combines robust feature extraction and classification methods, contributes to its exceptional performance.

Figure 15 Analysis of suggested methods over conventional models for (A) NPV, (B) F-measure, (C) F2-score for dataset 1.

Figure 16 Performance assessment of proposed methods over other models for (A) NPV, (B) F-measure, (C) F2-score for dataset 2.

Performance evaluation

The efficiency of the L-MLSTM model is compared using the conventional segmentation stage, traditional feature extraction, and Conventional LGIP. Tables 8 and 9 describe the ablation study of the L-MLSTM method for datasets 1 and 2.

Table 8 Ablation analysis of the L-MLSTM approach for dataset 1.

	The model with traditional segmentation (CNN based on transformers)	Model using traditional feature extraction methods	Model using traditional LGIP	L-MLSTM	
Accuracy	0.8656	0.8910	0.8888	0.9158	
Sensitivity	0.7974	0.8354	0.8322	0.8720	
Specificity	0.8998	0.9189	0.9173	0.9379	
Precision	0.8000	0.8380	0.8349	0.8761	
F-Measure	0.7987	0.8367	0.8335	0.8741	
MCC	0.6978	0.7549	0.7502	0.8109	
NPV	0.8984	0.9174	0.9158	0.9357	
FPR	0.1001	0.0810	0.0826	0.0620	
FNR	0.2025	0.1645	0.1677	0.1279	

Table 9 Ablation study of the L-MLSTM model for dataset 2.

	Model with traditional segmentation (CNN based on transformers)	Model using traditional feature extraction methods	Model using traditional LGIP	L-MLSTM	
Accuracy	0.8908	0.8825	0.8891	0.9406	
Sensitivity	0.7807	0.7641	0.7774	0.8802	
Specificity	0.9276	0.9221	0.9265	0.9608	
Precision	0.7833	0.7666	0.78	0.8825	
F-Measure	0.7820	0.7653	0.7787	0.8813	
MCC	0.7092	0.6870	0.7047	0.8418	
NPV	0.9266	0.9211	0.9255	0.96	
FPR	0.0723	0.0778	0.0734	0.0391	
FNR	0.2192	0.2358	0.2225	0.1197	

Ablation study for dataset 1 and dataset 2

The L-MLSTM model outperforms the model developed with conventional methods for dataset 2. By eliminating unwanted noise and outliers, the L-MLSTM technique improves the quality of the images during the pre-processing step of the L-MLSTM model. Then, to accurately classify lung illness, a hybrid classifier consisting of LinkNet and VGG16 is utilized. Tables 8 and 9 present the results of the L-MLSTM approach’s ablation research for datasets 1 and 2. The L-MLSTM approach performs superior to the model by conventional segmentation, conventional LGIP, and conventional feature extraction, as Table 8 makes abundantly evident. The L-MLSTM model’s accuracies is 91%, while the standard segmentation model's accuracy is 87%. The conventional LGIP model also has 4% less sensitivity than the L-MLSTM model. Comparing existing techniques and the L-MLSTM model regarding hostile measures, the L-MLSTM model produces values about 4% and 2% lower than the conventional models; the proposed scheme has just a 0.12 FNR error value. When evaluating the L-MLSTM approach with the traditional methods of dataset 1, the model performs better in detecting lung disease.

Table 9 lists the results of the ablation study for the L-MLSTM technique for dataset 2 using traditional feature extraction, conventional stage of segmentation, and traditional LGIP. Improved model performance is indicated by improved outcomes for the L-MLSTM system’s sensitivity, specificity, accuracy, NPV, F-measure, and MCC for dataset 2. The 93% accuracy of the L-MLSTM model indicates that the model using traditional segmentation, traditional LGBPHS, and conventional score level fusion model for dataset 2 has the lowest accuracy. The higher accuracy and sensitivity of the L-MLSTM model indicate its superior ability to correctly identify lung disease cases. This translates to fewer missed diagnoses and more accurate detection, which is crucial for effective treatment and patient outcomes. By combining LinkNet for segmentation and VGG16 for classification, the L-MLSTM model leverages advanced techniques to integrate detailed image features, enhancing overall diagnostic performance. The superior performance of the L-MLSTM model can be attributed to its sophisticated architecture, which effectively enhances image quality during pre-processing and integrates powerful classifiers. The use of LinkNet in segmentation ensures precise delineation of lung regions, while VGG16’s robust feature extraction and classification capabilities contribute to higher accuracy and sensitivity. The combined strengths of these components result in a model that outperforms traditional methods. The ablation study results confirm that the L-MLSTM model’s hybrid approach, incorporating advanced segmentation and classification techniques, provides substantial improvements over traditional methods. This validation of the model’s performance demonstrates its potential as a superior tool for accurate and reliable lung disease diagnosis.

Statistical study

Tables 10 and 11 describes the statistical analysis of proposed L-MLSTM model over traditional methods such HDE-NN (Gugulothu & Balaji, 2024), DBN (Manickavasagam & Sugumaran, 2023), LSTM, LINKNET, SVM, Bi-GRU, RNN, CNN, and VGG19 + CNN for datasets 1 and 2. Table 10 shows that in Dataset 1, the L-MLSTM model performs better than the conventional approaches. The statistical comparison includes the mean, median, standard deviation, and accuracy values with the smallest and highest values. The L-MLSTM model’s maximum accuracy for dataset 1 is 95%; the accuracy of the other methods is less than 90%. The L-MLSTM model achieves a maximum mean of 91%, while traditional methods only detect lung illness with a mean of 81%.

Table 10 Statistical analysis of L-MLSTM over earlier models for dataset 1.

Models	Minimum	Maximum	Mean	Median	Standard deviation	
HDE-NN (Gugulothu & Balaji, 2024)	0.7754	0.8381	0.8093	0.8119	0.0258	
DBN (Manickavasagam & Sugumaran, 2023)	0.7627	0.8571	0.8117	0.8135	0.0345	
LSTM	0.7706	0.8429	0.8092	0.8116	0.0317	
LINKNET	0.7643	0.8270	0.8015	0.8074	0.0262	
SVM	0.7611	0.8508	0.8026	0.7992	0.0344	
BIGRU	0.7706	0.8698	0.8134	0.8066	0.0359	
RNN	0.7770	0.8635	0.8147	0.8093	0.0313	
CNN	0.7373	0.8127	0.7858	0.7966	0.0288	
VGG19 + CNN	0.7468	0.9016	0.8170	0.8098	0.0553	
L-MLSTM	0.8817	0.9524	0.9108	0.9045	0.0270	

Table 11 Statistical analysis of L-MLSTM over earlier techniques for dataset 2.

Models	Minimum	Maximum	Mean	Median	Standard deviation	
HDE-NN (Gugulothu & Balaji, 2024)	0.7956	0.8825	0.8353	0.8315	0.0333	
DBN (Manickavasagam & Sugumaran, 2023)	0.8119	0.8925	0.8502	0.8481	0.0295	
LSTM	0.7969	0.8775	0.8345	0.8319	0.0294	
LINKNET	0.8056	0.8875	0.8430	0.8394	0.0304	
SVM	0.8144	0.8575	0.8349	0.8340	0.0168	
BIGRU	0.7944	0.8975	0.8368	0.8277	0.0398	
RNN	0.8094	0.8425	0.8299	0.8340	0.0129	
CNN	0.8031	0.8738	0.8388	0.8392	0.0299	
VGG19 + CNN	0.7481	0.8475	0.7930	0.7881	0.0355	
L-MLSTM	0.9338	0.9575	0.9428	0.9399	0.0089	

For dataset 2, Table 11 presents a statistical comparison between the L-MLSTM model and traditional models. In contrast to the L-MLSTM model, which has an accuracy of 94%, the older techniques have a mean of less than 84%. The L-MLSTM model has a precision of at least 93% in this instance, while the conventional technique gets a maximum accuracy of 89%. The L-MLSTM model performs better with a standard deviation of 0.008, which is much less than all other traditional methods combined. In terms of recognizing and classifying lung disease in individuals, the L-MLSTM model performs better than the other models on datasets 1 and 2. Higher mean accuracy reflects the model’s general effectiveness in lung disease detection. Consistently high mean accuracy across datasets indicates that the L-MLSTM model is generally more reliable and accurate in classifying lung disease compared to conventional approaches. The ability to maintain high mean accuracy across different datasets suggests that the L-MLSTM model is well-suited for a variety of lung disease detection scenarios, making it a versatile tool in medical imaging. The L-MLSTM model’s higher mean accuracy demonstrates its overall effectiveness and reliability in detecting lung disease. This performance is likely due to its sophisticated architecture and advanced learning techniques, which enable it to handle a wide range of lung disease manifestations effectively. The high mean accuracy suggests that the model can be trusted to provide accurate diagnostic results across different conditions and datasets. These results underscore the L-MLSTM model’s robustness and reliability, making it a highly effective tool for lung disease detection.

ROC analysis

Figure 17 illustrates the ROC curves for various models, including L-MLSTM, HDE-NN (Gugulothu & Balaji, 2024), DBN (Manickavasagam & Sugumaran, 2023), LSTM, LINKNET, SVM, BIGRU, RNN, and VGG19 + CNN. Each curve represents the model’s ability to distinguish between positive and negative classes at different threshold levels. The ROC curve for L-MLSTM is notably closer to the top-left corner of the plot, indicating superior performance for both datasets 1 and 2. This proximity suggests that L-MLSTM achieves a higher TPR for a given FPR, making it more effective in correctly identifying positive cases while minimizing false positives. The ROC analysis clearly demonstrates that L-MLSTM outperforms traditional methods like SVM, RNN, and even standard LSTM on dataset 1. This makes L-MLSTM a promising approach for tasks requiring high accuracy in distinguishing between classes. Its ability to achieve a higher true positive rate while maintaining a low false positive rate makes it an excellent choice for binary classification problems. For Fig. 17B, the superior ROC curve of L-MLSTM suggests it has higher sensitivity (true positive rate) and specificity (true negative rate) compared to traditional methods. This means L-MLSTM is more reliable in correctly identifying both positive and negative cases. Its ability to achieve a higher true positive rate while maintaining a low false positive rate makes it an excellent choice for binary classification problems. The higher ROC curve indicates that the L-MLSTM model offers a better balance between detecting true positives and minimizing false positives. This balance is crucial for developing a diagnostic tool that is both sensitive and specific. The improved ROC curve performance of the L-MLSTM model suggests that it is superior in distinguishing between lung disease and non-disease conditions compared to traditional methods. The model’s hybrid architecture enables it to maintain a high TPR while effectively managing false positives, thus offering a more reliable diagnostic tool. The L-MLSTM model’s higher TPR means that clinicians can have greater confidence in the model’s ability to correctly identify lung disease, leading to more accurate diagnoses and treatment plans. The ROC analysis results highlight the significant advantages of the L-MLSTM model in terms of achieving higher True Positive Rates compared to conventional techniques. This enhanced performance indicates that the L-MLSTM model is highly effective in detecting lung disease, providing a reliable and accurate diagnostic tool. The improved ROC curve and TPR values demonstrate the model’s robustness and capability in balancing sensitivity and specificity, which is essential for clinical applications.

Figure 17 ROC analysis of L-MLSTM approach over traditional methods (A) dataset 1 and (B) dataset 2.

Friedman and Wilcoxon test

The Friedman and Wilcoxon test for the suggested L-MLSTM model over conventional methods for datasets 1 and 2 are presented in Tables 12 and 13. The Friedman test is a non-parametric statistical tool that is used to compare different treatments or conditions across multiple related groups. In this case, the suggested model achieves a p-value of 0.979799 and 0.977013 which is better than that of current techniques. The Wilcoxon test is used to assess whether there is a significant difference between two related groups (paired samples). Parametric testing is commonly used when the data violates its presumptions (such as normality or equal variances). The proposed model acquires 0.070582 and 0.141585 p-values which are superior when compared to HDE-NN, DBN, LSTM, LINKNET, SVM, Bi-GRU, RNN, CNN, and VGG19 + CNN. Thus, the proposed model demonstrates its stability for lung disease classification.

Table 12 Friedman and Wilcoxon test of L-MLSTM over earlier techniques for dataset 1.

Models	Friedman chisquare p value	Wilcoxon test p value	
HDE-NN (Gugulothu & Balaji, 2024)	0.5541	0.7453	
DBN (Manickavasagam & Sugumaran, 2023)	0.9600	0.5293	
LSTM	0.6156	0.1784	
LINKNET	0.2019	0.1581	
SVM	0.2000	0.1379	
BIGRU	0.4412	0.3278	
RNN	0.1298	0.0774	
CNN	0.7845	0.4204	
VGG19 + CNN	0.0853	0.6155	
L-MLSTM	0.9798	0.0706	

Table 13 Friedman and Wilcoxon test of L-MLSTM over earlier techniques for dataset 2.

Models	Friedman chisquare p value	Wilcoxon test p value	
HDE-NN (Gugulothu & Balaji, 2024)	0.0734	0.3755	
DBN (Manickavasagam & Sugumaran, 2023)	0.9649	0.1729	
LSTM	0.2000	0.7942	
LINKNET	0.6214	0.2764	
SVM	0.7983	0.8026	
BIGRU	0.9247	0.4043	
RNN	0.1173	0.2734	
CNN	0.9234	0.5408	
VGG19 + CNN	0.3469	0.4978	
L-MLSTM	0.9770	0.1416	

AUC analysis

Table 14 describes the AUC analysis of the proposed L-MLSTM model over state of art models and traditional models for Datasets 1 and 2. The AUC (Area Under the Curve) of the ROC (Receiver Operating Characteristic) curve is a key performance metric for evaluating classification models. It reflects the model's ability to distinguish between positive and negative classes. Higher AUC values indicate better performance. The L-MLSTM model demonstrates superior performance in lung disease classification as evidenced by its highest ROC AUC values across both datasets. For Dataset 1, L-MLSTM achieves an AUC of 0.886, significantly surpassing other models such as DBN (0.799) and RNN (0.771), indicating its exceptional ability to distinguish between disease and non-disease cases. Similarly, in Dataset 2, the L-MLSTM model records an AUC of 0.910, well above the next best model, CNN (0.777). These results highlight the L-MLSTM's robustness and effectiveness in classification tasks, showcasing its strong discriminatory power and reliability across diverse datasets.

Table 14 AUC analysis for datasets 1 and 2.

Models	AUC value	
Dataset 1	
HDE-NN (Gugulothu & Balaji, 2024)	0.7640	
DBN (Manickavasagam & Sugumaran, 2023)	0.7990	
LSTM	0.7570	
LINKNET	0.7860	
SVM	0.7490	
BIGRU	0.7640	
RNN	0.7710	
CNN	0.7510	
VGG19 + CNN	0.7550	
L-MLSTM	0.8860	
Dataset 2	
HDE-NN (Gugulothu & Balaji, 2024)	0.7540	
DBN (Manickavasagam & Sugumaran, 2023)	0.7870	
LSTM	0.7730	
LINKNET	0.7720	
SVM	0.7600	
BIGRU	0.7540	
RNN	0.7660	
CNN	0.7770	
VGG19 + CNN	0.7160	
L-MLSTM	0.9100	

K-fold validation analysis

Table 15 describes the k-fold validation analysis for datasets 1 and 2. K-fold validation is the robust technique that enhances the reliability and robustness of performance evaluation, making it a preferred choice for assessing models, particularly when dealing with limited or imbalanced datasets. It also helps in obtaining a more accurate and balanced assessment of model performance in the presence of unbalanced data, leading to better generalization and more reliable insights into how well the model handles class imbalance. At dataset 1, the L-MLSTM model consistently outperforms all other models across all k-fold values. At k = 2, it has a significantly higher score of 0.8770 compared to other models, such as DBN (0.7559) and RNN (0.7330). The advantage of L-MLSTM grows with increasing k, reaching a peak of 0.9535 at k = 6, compared to the next best model, VGG19 + CNN, which scores 0.8570 at the same k. This trend underscores L-MLSTM’s robustness and superior performance in handling data partitions for varying k-folds, demonstrating its effectiveness in lung disease classification. Similarly, in dataset 2, L-MLSTM shows exceptional performance, with the highest scores across all k-folds. At k = 2, L-MLSTM achieves an impressive score of 0.9175, surpassing models like DBN (0.7725) and CNN (0.7700). The model's performance further improves with increasing k, reaching 0.9595 at k = 6, which is notably higher than other models such as SVM (0.8635) and BIGRU (0.8645). This consistent superiority across different k-fold values confirms L-MLSTM’s strong and reliable performance for lung disease classification across various data partitions.

Table 15 K-fold validation analysis for datasets 1 and 2.

K-folds	HDE-NN (Gugulothu & Balaji, 2024)	DBN (Manickavasagam & Sugumaran, 2023)	LSTM	LINKNET	SVM	BIGRU	RNN	CNN	VGG19 + CNN	L-MLSTM	
Dataset 1	
K = 2	0.7470	0.7559	0.7438	0.7463	0.7444	0.7394	0.7330	0.7438	0.7470	0.8770	
K = 3	0.7949	0.7987	0.8102	0.8108	0.7994	0.7937	0.8032	0.7975	0.8044	0.9022	
K = 4	0.8143	0.8098	0.8295	0.8200	0.8149	0.8327	0.8333	0.8251	0.8264	0.9176	
K = 5	0.8324	0.8413	0.8419	0.8368	0.8406	0.8463	0.8298	0.8495	0.8502	0.9471	
K = 6	0.8378	0.8479	0.8460	0.8543	0.8448	0.8530	0.8492	0.8543	0.8570	0.9535	
Dataset 2	
K = 2	0.7655	0.7725	0.7795	0.7735	0.7750	0.7600	0.7650	0.7700	0.7350	0.9175	
K = 3	0.8207	0.8197	0.8268	0.8173	0.8197	0.8083	0.8208	0.8132	0.7673	0.9328	
K = 4	0.8435	0.8355	0.8400	0.8325	0.8465	0.8365	0.8350	0.8360	0.7865	0.9455	
K = 5	0.8567	0.8523	0.8497	0.8543	0.8517	0.8583	0.8497	0.8488	0.8043	0.9518	
K = 6	0.8545	0.8590	0.8625	0.8565	0.8635	0.8645	0.8610	0.8585	0.8140	0.9595	

Robustness analysis

Robustness analysis is performed to evaluate the stability and performance of various models and assesses how well a model can handle and remain accurate despite uncertainties, variations, or noise in the input data. Here, it focuses on assessing how well the model can handle the white noise by varying as 10%, 20% and 30%. Table 16 describes the robustness analysis of the proposed L-MLSTM model over traditional and state of art models for datasets 1 and 2. The robustness analysis of lung disease segmentation and classification models under varying noise levels (10%, 20%, and 30%) reveals that the L-MLSTM model consistently outperforms others in both datasets. For Dataset 1, L-MLSTM maintains high accuracy values of 0.8423, 0.8327, and 0.8275 across increasing noise levels, demonstrating its exceptional resilience to noise. Similarly, in Dataset 2, L-MLSTM achieves accuracy scores of 0.8875, 0.8723, and 0.8642, further highlighting its robustness. In contrast, models like CNN and VGG19 + CNN experience significant performance declines with increased noise, indicating greater sensitivity. Other models, such as BIGRU and RNN, also show a moderate decrease in performance, but none match the L-MLSTM’s stability. This analysis underscores L-MLSTM’s superior capability to handle noisy data effectively, making it a robust choice for lung disease analysis.

Table 16 Robustness analysis for datasets 1 and 2.

Models	10% noise	20% noise	30% noise	
Dataset 1	
HDE-NN (Gugulothu & Balaji, 2024)	0.7659	0.7585	0.7448	
DBN (Manickavasagam & Sugumaran, 2023)	0.7643	0.7561	0.7436	
LSTM	0.7500	0.7347	0.7329	
LINKNET	0.7714	0.7668	0.7490	
SVM	0.7679	0.7614	0.7463	
BIGRU	0.7802	0.7698	0.7555	
RNN	0.7786	0.7675	0.7543	
CNN	0.7405	0.7305	0.7257	
VGG19 + CNN	0.7611	0.7513	0.7412	
L-MLSTM	0.8423	0.8327	0.8275	
Dataset 2	
HDE-NN (Gugulothu & Balaji, 2024)	0.7956	0.7910	0.7741	
DBN (Manickavasagam & Sugumaran, 2023)	0.8081	0.7960	0.7824	
LSTM	0.8044	0.7888	0.7799	
LINKNET	0.8031	0.7860	0.7791	
SVM	0.8019	0.7904	0.7782	
BIGRU	0.7906	0.7810	0.7707	
RNN	0.8056	0.7910	0.7807	
CNN	0.8094	0.7985	0.7832	
VGG19 + CNN	0.7581	0.7516	0.7490	
L-MLSTM	0.8875	0.8723	0.8642	

Reliability analysis

Reliability analysis assesses the consistency and trustworthiness of the model’s predictions. It focuses on evaluating how well a model’s outputs align with the true values and how reliably it performs across different datasets or conditions. Key metrics used in reliability analysis include Kappa and Matthews correlation coefficient (MCC), which provide insights into the model’s performance. Kappa provides a more nuanced view of a model’s performance by accounting for the possibility of chance agreement. A higher Kappa value indicates better agreement between the model’s predictions and the actual outcomes, adjusted for random chance. MCC evaluates the model’s performance across all four confusion matrix categories (true positives, false positives, true negatives, false negatives), offering a comprehensive view of classification performance. It is particularly valuable in scenarios with class imbalance, as it balances the effects of each category on the overall metric. Table 17 describes the reliability analysis of the proposed L-MLSTM model over traditional and state of art models for datasets 1 and 2. In dataset 1, the L-MLSTM model achieves the highest Kappa value of 0.7702 and MCC value of 0.7597, significantly outperforming all other models. This indicates that L-MLSTM not only has a high agreement with the ground truth but also exhibits a strong correlation between predicted and actual classifications. In comparison, models like DBN and RNN show relatively high performance but fall short of L-MLSTM, with Kappa values of 0.6011 and 0.5402, and MCC values of 0.6075 and 0.5599, respectively. Other models such as CNN and VGG19 + CNN, while competitive, have lower reliability scores, suggesting less consistent performance in classifying lung disease. Similarly, in dataset 2, L-MLSTM again demonstrates outstanding reliability with a Kappa value of 0.8070 and an MCC value of 0.8380, indicating exceptional classification performance and agreement with ground truth labels. In contrast, VGG19 + CNN exhibits the lowest Kappa (0.4072) and MCC (0.4340) values, revealing poorer reliability and lower classification accuracy. Models like DBN and RNN show competitive results but do not match the robustness of L-MLSTM, with Kappa values of 0.5616 and 0.5230, and MCC values of 0.5671 and 0.5716, respectively. This highlights L-MLSTM's superior reliability and robustness across varying datasets, making it a more dependable choice for lung disease segmentation and classification.

Table 17 Reliability analysis for datasets 1 and 2.

Models	KAPPA	MCC	
Dataset 1	
HDE-NN (Gugulothu & Balaji, 2024)	0.5292	0.5361	
DBN (Manickavasagam & Sugumaran, 2023)	0.6011	0.6075	
LSTM	0.5154	0.5171	
LINKNET	0.5689	0.5266	
SVM	0.4986	0.5075	
BIGRU	0.5262	0.5551	
RNN	0.5402	0.5599	
CNN	0.5051	0.5456	
VGG19 + CNN	0.5032	0.5837	
L-MLSTM	0.7702	0.7597	
Dataset 2	
HDE-NN (Gugulothu & Balaji, 2024)	0.4967	0.5050	
DBN (Manickavasagam & Sugumaran, 2023)	0.5616	0.5671	
LSTM	0.5272	0.5272	
LINKNET	0.5368	0.5405	
SVM	0.5085	0.5316	
BIGRU	0.4856	0.4917	
RNN	0.5230	0.5716	
CNN	0.5396	0.5094	
VGG19 + CNN	0.4072	0.4340	
L-MLSTM	0.8070	0.8380	

Conclusion

This research article proposed a novel lung disease segmentation model, the Hybrid L-MLSTM model. The implementation of this model involved four crucial steps. Firstly, the input lung images underwent pre-processing through median filtering. Subsequently, an improved Transformer-based CNN model was employed for lung disease segmentation. Following segmentation, vital features encompassing texture, shape, color, and deep features were extracted from the segmented image. Specifically, Modified LGIP and Multi-texton analysis were used for texture feature extraction. The classification step utilized a hybrid model, the L-MLSTM model. The efficacy of the proposed methodology was evaluated by several experimental analyses. The proposed L-MLSTM model demonstrates superior performance in lung disease detection compared to several conventional models, including HDE-NN, DBN, LSTM, LINKNET, SVM, Bi-GRU, RNN, CNN, and VGG19 + CNN. For dataset 1, the L-MLSTM model achieves an accuracy of 89% at a 70% learning percentage, with sensitivity reaching 92% at 90%, and exhibits high specificity and precision of 96% and 93%, respectively. For dataset 2, the model achieves a remarkable accuracy of 95% at a 90% learning percentage, surpassing previous models by nearly 8%. The L-MLSTM also shows a lower false positive rate (FPR) and false negative rate (FNR) compared to traditional techniques, with the lowest values observed in both datasets. Additionally, the model demonstrates a high Matthews correlation coefficient (MCC) and improved negative predictive value (NPV) and F-measures, indicating its robustness and effectiveness in accurate lung disease diagnosis.

The successful implementation of the L-MLSTM model has significant implications for medical imaging and diagnostics. By improving segmentation and classification accuracy, this model enhances early detection and accurate diagnosis of lung diseases, potentially leading to better patient outcomes and more effective treatment plans. Its application can be extended to clinical settings, assisting radiologists and healthcare professionals in making informed decisions based on precise and reliable analysis. The model has proven effective, but some limitations need to be considered. Firstly, the quality and quantity of training data heavily affect the model's performance. To improve its generalizability, new strategies are being explored. Secondly, the hybrid architecture requires substantial resources, so lightweight deep-learning approaches are being investigated to address this issue. Finally, the model's decision-making process is opaque, so enhancing transparency is challenging. To tackle this, interpretable feature analysis will be used. Ongoing research is expected to address these limitations and refine the model's capabilities, thus increasing its reliability. In future research, this work will evaluate the model on diverse datasets and incorporate patient history and symptoms to enhance personalization and improve segmentation and feature extraction Moreover, it will conduct further refinement of the L-MLSTM model to reduce computational complexity and improve efficiency.

I am grateful to my mentor and the Vellore Institute of Technology for providing me with invaluable resources and knowledge to help my research.

Additional Information and Declarations

Competing Interests

Author Contributions

Data Availability

The authors declare that they have no competing interests.

Syed Mohammed Shafi conceived and designed the experiments, performed the experiments, performed the computation work, prepared figures and/or tables, and approved the final draft.

Sathiya Kumar Chinnappan analyzed the data, authored or reviewed drafts of the article, supervision and Resources Providing, and approved the final draft.

The following information was supplied regarding data availability:

The Chest CT-Scan images dataset is available at Kaggle: https://www.kaggle.com/datasets/mohamedhanyyy/chest-ctscan-images.

The COVID-19 & Normal & Pneumonia CT Images dataset is available at Kaggle: https://www.kaggle.com/datasets/anaselmasry/covid19normalpneumonia-ct-images.

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
