# Peer review of "Hybrid transformer-CNN and LSTM model for lung disease segmentation and classification"

_PeerJ Computer Science, doi:10.7717/peerj-cs.2444_

## Round 0.1 · original submission · Major Revisions

Dear authors,

Thank you for submitting your article. Reviewers have now commented on your article and suggest major revisions. We do encourage you to address the concerns and criticisms of the reviewers and resubmit your article once you have updated it accordingly.

When submitting the revised version of your article, it will be better to address the following:

1. The abstract does not present the creation or usage of the dataset.
2. Figures should be polished. The word error in the figures (for example Figure 4, Figure 5, etc.) created by taking screenshots needs to be corrected.
3. All reported graphs should be accompanied by some concrete description of the lessons learned from the results reflected in the graph. It is important to explain them in detail and to enrich them with some semantics by showing the reasons for these results, how they can be further improved, etc.
4. Explanation of the equations should be checked. All variables should be written in italic as in the equations. Their definitions and boundaries should be defined. Necessary references should also be given.
5. English grammar and writing style errors should be corrected.

Best wishes,

Reviewer 1 ·

Basic reporting

The authors in this paper titled “Improved Transformer-CNN model-based Lung segmentation with Hybrid Classification 2 Model: Combining Shape, Texture, Color, and Deep Feature set” proposed a lung disease segmentation with a hybrid LinkNet-Modified LSTM (L-MLSTM) model and highlighted four of steps of the overall process.

The authors are suggested to address these comments while revising the paper.

Title:
The authors are suggested to revise the title as “Hybrid Transformer-CNN and LSTM Model for Lung Disease Segmentation and Classification”.

Abstract:
The authors are suggested to revise the abstract to include details on the evaluation methods used to assess the model's performance, specific findings and results of the study, metrics such as accuracy, precision, recall, F1 scores, and a discussion on the significance of the results in the context of lung disease diagnosis and how the proposed model improves upon existing methods.

Rewrite and improve the research contributions listed at the end of the introduction section. They should clearly outline the novel aspects of the proposed model, its expected impact on lung disease diagnosis and classification, and how it addresses gaps in the existing literature.

Table 1 provided at the end of the literature review is not exhaustive. Presently, it includes only 7 studies. The authors are suggested to improve the literature review section by adding more recent studies from 2024 and then enhance Table 1 as a summary of the literature review to ease readers by providing a quick overview of the summary in tabular format and highlighting the existing methodologies, datasets, findings, and gaps.

A. Problem Definition:
This is not defining the problem as it is not being defined or formulated. The authors should add this as the last paragraph of the literature review that actually highlights and summarizes the research gaps in the given literature,

Experimental design

Redraw Figure 1 in a more proper and detailed way and also improve the caption instead of using words like "Diagrammatic representation".

The paper does not provide enough detail about the implementation of the L-MLSTM model. It's crucial to describe the architecture, the parameters used, and the specific modifications made to the LSTM model to ensure reproducibility.

There is no clear justification for choosing the hybrid L-MLSTM model over other possible models. The authors should explain why this particular model was selected and how it compares to other state-of-the-art models.

More information is needed on the pre-processing steps and the rationale behind choosing these specific datasets. Are these datasets publicly available? How the datasets are collected?

The paper should describe how the data was split into training, validation, and test sets to avoid data leakage and ensure fair evaluation.

The paper mentions using different evaluation metrics but does not provide a rationale for selecting these metrics.

The experimental design should include comparisons with baseline models to highlight the improvements brought by the proposed model. Also if the datasets are publicly available and being used by previous studies also provides a comparison.

Validity of the findings

The authors are suggested to include a detailed discussion on how their findings advance current knowledge and validate their results with additional experiments or comparisons to enhance reliability and robustness.

Improve the conclusion section and describe the key achievements of the research, limitations, implications, and future directions. Do not include limitations as a separate section.

Additional comments

Improve the quality of all figures and tables.

Reviewer 2 ·

Basic reporting

1. The authors needs to work on the grammatical errors.

Experimental design

1. It is advisable to the authors to clearly depict which knowledge gap they are targeting and how their proposed research is effectively filling it.
2. Also, please include the experimental setup used for the proposed research.

Validity of the findings

There is no comparison done with the any State-Of-Art methodologies present in the literature specifically proposed for the chosen research problem. Please include few of them otherwise how would the readers come to know that there is no biasness is there in your paper and your proposed methodology has surpassed other proposed techniques. And this comparison should be done in tabular form after applying them on your datasets or applying your methodology for the dataset given in them.

Additional comments

1. Please include some quantitative results in the abstract as it will help in arising reader’s interest to read further.
2. In Section 1 of Introduction, it is not effectively communicating the purpose and background of the chosen research problem. Please carefully include why there is need of your proposed methodology.
3. In Section 2 of Related Works, please include less details of the papers and include what are the research gaps in the present literature. If the readers want to read further, they can go to the cited reference.
4. There is very less information and discussion about the novelty proposed in the manuscript. Please focus only on the novelty proposed if any.
5. There are already many techniques present in the literature for the chosen research problem. So, it is advisable to the authors to clearly depict the USP and significance of your research work for the chosen research problem.

Reviewer 3 ·

Basic reporting

The problem that the authors are trying to solve is very interesting. They proposed a novel lung disease segmentation model, the Hybrid L-MLSTM model, which involved four crucial steps. Firstly, the input lung images underwent pre-processing through median filtering. Subsequently, an improved Transformer-based CNN model was employed for lung disease segmentation. Following segmentation, vital features encompassing texture, shape, color, and deep features were extracted from the segmented images. Specifically, Modified LGIP and Multi-texton analysis were used for texture feature extraction. The classification step utilized a hybrid model, the L-MLSTM model. However, I suggest the authors update a few things in terms of the format and content so that readers can more easily understand their motivations and their experimental and theoretical results.



General comments about the format:

- All mathematical equations and formulas should be rewritten in more clear way, Please make sure that words such as (if, otherwise, ...) in equations, are not written in the same format as the mathematical equations characters and that there is a space between words and equations.

- Figures should reformatted and cleaned (Figures 4 and 5, ...),

- Results figures are not readable (Figures 8 and 9 )

- Figures should not reshaped manually (Figures 10, 11, 12).

- The indentation of all sections should be corrected, and subsections should indented the same way within one section.

- Some references links are not working, or not placed on the right place, for example reference [43] is is mentioned in line 468, near to the description of comb-H-sine function and leaky Relu, but the link refers to a kaggle dataset url that is not working.

- Another example of the references that should be updated, reference [40] it is support to describe VGG16 model architecture, but the article in reference [40] does not mention anything about VGG16.

Experimental design

I understand that you would like to enhance non-linearities introduced into the model, but it is not well explained whsy `ScRelu` is a good activation function for your use case. There are plenty of non-linear activation functions that can enhance non-linearities.

- Why do you name it as `ScRelu`? Readers may be confused with Scaled Rectified Linear Unit.
- What are the properties of the `ScRelu` that makes it a good candidate ?
- I think it would be great if you can add a unidimensional plot of this new activation function.
- Equations that describe this activation function are not clear, especially equation 22.
- There is a typo in your comb-H-sine function, do you mean sinh^(-1) instead of sin^(-1) in line 468, eq 20 and 22,
- What do you mean by sinh^(-1)? If you mean the primitive of sinh you can write directly cosh, if you mean 1/sinh, it would better if you write it as a fraction, to not confuse the reader.
- can you please explain what are the benefits to sum up a bounded function (sigmoid) with unbounded function com-H-sine, near to +/- infinity the sigmoid will not have any effect of the activation function ?

Validity of the findings

- The confusion matrices in figures (8, 9, ) are not clear and was not able to read them

- You focused on the accuracy metric to evaluate the performance of your models, and this metric is not the best metric to consider in cases such the lung disease segmentation and classification. I think it is essential to focus more and evaluate your models using recall and F2 score, AUC. Furthermore, the accuracy is not representative in scenarios where we have unbalanced data.

- You also did a ROC analysis but I think it could be improved since, you used only one data point to plot the the ROC curves, I think it is important to use different data points to draw to ROC curves, we cannot compare the ROC curves of different models using one data point, using multiple data points to draw the ROC curves may give different results.

---

## Round 0.2 · accepted · Accept

Dear authors,

One of the original reviewers did not respond to the invitation for reviewing your revised manuscript. One reviewer accepted the invitation however exceeded the time limit for uploading the review. The other reviewer thinks your paper can be accepted. I also think that the paper has been sufficiently improved. As such, the article is considered acceptable.

Best wishes,

Reviewer 1 ·

Basic reporting

No Comment

Experimental design

No Comment

Validity of the findings

No Comment